# A review of the (Revised) Universal Soil Loss Equation (R/USLE): with a view to increasing its global applicability and improving soil loss estimates

Rubianca Benavidez[1], Bethanna Jackson[1], Deborah Maxwell[1], Kevin Norton[1]

[1]School of Geography, Environment, and Earth Sciences, Victoria University of Wellington, Wellington, 6012, New Zealand

*Correspondence to*: Rubianca Benavidez (Rubianca.Benavidez@vuw.ac.nz)

**Abstract.** Soil erosion is a major problem around the world because of its effects on soil productivity, nutrient loss, siltation in water bodies, and degradation of water quality. By understanding the driving forces behind soil erosion, we can more easily identify erosion-prone areas within a landscape to address the problem strategically. Soil erosion models have been used to assist in this task. One of the most commonly used soil erosion models is the Universal Soil Loss Equation (USLE) and its family of models: the Revised Universal Soil Loss Equation (RUSLE), the Revised Universal Soil Loss Equation version 2 (RUSLE2), and the Modified Universal Soil Loss Equation (MUSLE). This paper reviewed the different sub-factors of USLE and RUSLE, and analysed how different studies around the world have adapted the equations to local conditions. We compiled these studies and equations to serve as a reference for other researchers working with R/USLE and related approaches. Within each sub-factor section, the strengths and limitations of the different equations are discussed and guidance is given as to which equations may be most appropriate for particular climate types, spatial resolution, and temporal scale. We investigate some of the limitations of existing R/USLE formulations, such as uncertainty issues given the simple empirical nature of the model and many of its subcomponents, uncertainty issues around data availability, and its inability to account for soil loss from gully erosion, mass wasting events, or predicting potential sediment yields to streams. Recommendations on how to overcome some of the uncertainties associated with the model are given. Several key future directions to refine it are outlined: e.g. incorporating soil loss from other types of soil erosion, estimating soil loss at sub-annual temporal scales, and compiling consistent units for future literature to reduce confusion and errors caused by mismatching units. The potential of combining R/USLE with the Compound Topographic Index (CTI) and Sediment Delivery Ratio (SDR) to account for gully erosion and sediment yield to streams respectively is discussed. Overall, the aim of this paper is to review the R/USLE, its sub-factors, and to elucidate the caveats, limitations, and recommendations for future applications of these soil erosion models. We hope these recommendations will help researchers more robustly apply R/USLE in a range of geoclimatic regions with varying data availability, and modelling different land cover scenarios at finer spatial and temporal scales (e.g. at the field scale with different cropping options).

# 1 Introduction

Soil erosion involves many processes but the overall effect is of particles being transported and deposited from one location to another. Although it occurs naturally, soil erosion is often exacerbated by anthropogenic activities (Adornado et al., 2009). Soil erosion is affected by wind, rainfall and associated runoff processes, vulnerability of soil to erosion, and the

characteristics of land cover and management (David, 1988; Aksoy & Kavvas, 2005, Panagos et al., 2015e). Managing and understanding erosion and associated degradation is critical because of its possible effects: nutrient loss, river and reservoir siltation, water quality degradation, and decreases in soil productivity (Bagherzadeh, 2014). In a review of the costs of soil erosion, Pimentel et al. (1995) reported soil erosion rates for regions around the world: Asia, South America, and Africa with an average of 30 to 40 ton ha$^{-1}$ yr$^{-1}$ and an average of 17 ton ha$^{-1}$ yr$^{-1}$ for the United States of America and Europe. For

comparison, the soil erosion rate for undisturbed forests was reported to range from 0.004 ton ha$^{-1}$ yr$^{-1}$ to 0.05 ton ha$^{-1}$ yr$^{-1}$ globally (Pimentel et al., 1995). Within a landscape, erosion due to water can be caused by unconcentrated flow (sheet), within small channels (rills), raindrop impact and overland flow (inter-rill), and larger channels of concentrated flow (gullies) (Aksoy & Kavvas, 2005; Morgan, 2005). Land management can be improved through understanding how these erosion processes occur and what areas are vulnerable to soil loss. Advances in technology such as the development of soil erosion models and

increases in computing power for spatial analysis have assisted in making soil erosion modelling faster and more accurate.

Soil erosion models aid land management by helping understand the areas vulnerable to soil erosion in the baseline scenario, potential erosion rates, and possible causes of soil erosion. They range from relatively simple empirical models, and conceptual models, to more complicated physics-based models (Merritt et al., 2003). Like any other model, there are uncertainties associated with soil erosion models that cannot account for all the complex interactions of sediment delivery.

Hence, unless extensive parameterisation and validation against observed data is accomplished, soil loss rates from models should be taken as best available estimates instead of absolute values (Wischmeier & Smith, 1978). Extensive reviews of soil erosion models of varying complexity have been done before but tend to focus on input requirements and applications (Aksoy & Kavvas, 2005; Merritt et al., 2003). A review by de Vente & Posen (2005) differs by focusing on semi-quantitative models that include different types of soil erosion in order to estimate basin sediment yield. Other reviews have focused on the use of

different types of soil erosion models in particular places, such as Brazilian watersheds for de Mello et al. (2016).

One family of empirical soil loss models is the Universal Soil Loss Equation (USLE) suite of models including the original USLE, the Revised Universal Soil Loss Equation (RUSLE), the Revised Universal Soil Loss Equation version 2 (RUSLE2), and the Modified Universal Soil Loss Equation (MUSLE). The USLE is an empirical model used to estimate the average rate of soil erosion (tons per unit area) for a given combination of crop system, management practice, soil type, rainfall

pattern, and topography. It was originally developed at the plot-scale for agricultural plots in the United States of America (Wischmeier & Smith, 1978). An updated form of USLE (RUSLE) was published to include new rainfall erosivity maps for the United States of America and improvements to the method of calculating the different USLE factors (Renard et al., 1997). RUSLE added changes in soil erodibility due to freeze-thaw and soil moisture, a method for calculating cover and management

factors, changes to how the influence of topography is incorporated into the model, and updated values to represent soil conservation practices (Renard & Freimund, 1994). The RUSLE2 framework is a computer interface to handle more complex field situations, including an updated database of factors (Foster et al., 2003). These three variations of R/USLE measure soil loss per unit area at an annual time scale. The MUSLE is an extension to work at finer temporal resolution, using runoff and peak flow rate to estimate event-based soil loss (Sadeghi et al., 2014). These models have been used around the world due to their relative simplicity and seemingly low data requirements (Table A1).

This simplicity of the R/USLE has been integrated into more complex soil erosion models to help with management and decision-making, including the Agricultural Non-Point Source model (AGNPS), the Chemical Runoff and Erosion from Agricultural Management Systems model (CREAMS), and the Sediment River Network model (SedNet) (Aksoy & Kavvas, 2005; de Vente & Poesen, 2005; Merritt et al., 2003). The AGNPS estimates upland erosion using the USLE and then uses sediment transport algorithms to simulate runoff, sediment and nutrient transport within watersheds (Aksoy & Kavvas, 2005). The usage of R/USLE in large models is mainly for the purpose of assisting with decision-making, such as prioritising land use objectives in the Philippines (Bantayan & Bishop, 1998), scenario analysis for water quality in catchments in New Zealand (Rodda et al., 2001), or delineating unique soil landscapes in Australia (Yang et al., 2007).

Extensive reviews of soil erosion modelling and types of soil erosion models have been published that briefly discuss the R/USLE as an empirical model, elements of which are commonly incorporated into more complex conceptual or physics-based soil erosion models (Aksoy & Kavvas, 2005; de Vente & Poesen, 2005; Merritt et al., 2003). This review is more specific to the R/USLE and addresses the complexity of its different subfactors, as well as the issues for researchers to consider before applying R/USLE to their study area. These issues range from equation choices, DEM resolution, granularity in land cover characteristics, scale, etc. The MUSLE is not included in this review because Sadeghi et al. (2014) have already done an extensive review of the model and event-scale estimates are beyond the scope of this paper. Annual estimates of soil loss are useful for understanding the baseline erosion in a catchment, but intra-annual and event-based soil loss estimates are useful to elucidate temporal variations in erosion. Performing event-based soil loss modelling is important for areas that frequently experience extreme events as these can cause large-scale sediment transport and mass wasting.

The main aim of this paper is to review the (Revised) Universal Soil Loss Equation and its sub-factors through the following objectives:

- Review the USLE and RUSLE literature to compile equations for the different sub-factors within the R/USLE;
- Provide guidance as to which datasets and equations are appropriate over a range of geoclimatic regions with varying levels of data availability;
- Outline the limitations and caveats of the R/USLE that future users must consider; and
- Outline potential future directions to overcome these limitations and to improve R/USLE applications

## 2 Universal Soil Loss Equation (USLE)

The principal equation for the USLE model family is below:

$$A = R \times K \times L \times S \times C \times P \quad (1)$$

Where:

| | |
|---|---|
| A | Mean annual soil loss (metric tons hectare$^{-1}$ year$^{-1}$) |
| R | Rainfall and runoff factor or rainfall erosivity factor (megajoules millimetre hectare$^{-1}$ hour$^{-1}$ year$^{-1}$) |
| K[1] | Soil erodibility factor (metric tons hour megajoules$^{-1}$ millimetre$^{-1}$) |
| L | Slope-length factor (unitless) |
| S | Slope-steepness factor (unitless) |
| C | Cover and management factor (unitless) |
| P | Support practice factor (unitless) |

The USLE was originally developed at the farm-plot scale for agricultural land in the United States of America, but has seen use in many other countries, scales, and geoclimatic regions. Although the name implies that the model can be applied to all soils, the original USLE is more accurate for soils with medium texture, slopes of less than 400ft in length with a gradient ranging between 3 and 18% and managed with consistent cropping practises that are well-represented in plot-scale erosion
studies (Wischmeier & Smith, 1978). Hence, applying the USLE family of models to soils and sites exceeding these limits requires careful parameterisation of the model and being mindful of the increased uncertainty in model predictions.

In the original development of the model, this farm plot is called the "unit plot" and is defined as a plot that is 22.1m long, 1.83m wide, and has a slope of 9% (Wischmeier & Smith, 1978). Although the model accounts for rill and inter-rill erosion, it does not account for soil loss from gullies or mass wasting events such as landslides (Thorne et al., 1985). The
appendix of this paper compiles a non-exhaustive list of studies that have applied the USLE and RUSLE models to watersheds around the world.  The uncertainties in soil erosion modelling stem from the availability of long-term reliable data, which includes issues of temporal resolution (e.g. <30-minute resolution required for R/USLE) and the availability of spatial data over a catchment. This issue is not unique to R/USLE applications and is more pressing for more complex models that have a large amount of variables that require detailed data (de Vente & Poesen, 2005; Hernandez et al., 2012). Hence, the ubiquitous
usage of the R/USLE can be attributed to its relatively low data requirements compared to more complex soil loss models, making it potentially easier to apply in areas with scarce data. Another limitation of the R/USLE and arguably many erosion model applications is the lack of validation data to verify model outputs, which is discussed further in Section 4.

Although the application of the R/USLE seem to be a simple linear equation at first glance, this review addresses the complex equations that go into calculating its factors, such as rainfall erosivity which requires detailed pluviographic data (<
30 minute resolution). This paper discusses the advantages, disadvantages, and limitations of the USLE model family.

---

[1] The RUSLE handbook by Renard et al. (1997) indicates that the K-factor metric units are metric tons hectare hour megajoules$^{-1}$ hectare$^{-1}$ millimetre$^{-1}$, but for mathematical correctness, the hectare units cancel out.

Although alternative equations are presented, we also discuss questions of suitability that future users should consider before applying the R/USLE.

## 2.1 Rainfall erosivity factor (R)

The R-factor represents the effect that rainfall has on soil erosion and was included after observing sediment deposits after an intense storm (Wischmeier & Smith, 1978). The annual R-factor is a function of the mean annual $EI_{30}$ that is calculated from detailed and long-term records of storm kinetic energy (E) and maximum thirty-minute intensity ($I_{30}$) (Morgan, 2005; Renard et al., 1997). Due to the detailed data requirements for the standard R/USLE calculation of rainfall erosivity, studies in areas with less detailed data have used alternative equations depending on the temporal resolution and availability of the rainfall data. These compiled studies have used long-term datasets with at least daily temporal resolution to construct their R-factor equation. Extensive work by Naipal et al. (2015) attempted to apply the R/USLE at a coarse global scale (30 arcsecond) by using USA and European databases to derive rainfall erosivity equations. These equations use a combination of annual precipitation (mm), mean elevation (m), and simple precipitation intensity index (mm day$^{-1}$) to calculate the R-factor for different Köppen-Geiger climate classifications (Naipal et al., 2015). Loureiro and Coutinho (2001) used 27 years of daily rainfall data from Portugal and the R/USLE method of calculating $EI_{30}$ to construct an equation that uses the number of days that received over 10.0 mm of rainfall and the amount of rainfall per month when the day's rainfall exceeded 10.0 mm. The Loureiro and Coutinho (2001) equation was modified by Shamshad et al. (2008) for use in tropical Malaysia by using long-term rainfall data to construct a regression equation relating monthly rainfall and annual rainfall with the R-factor. Similarly, Sholagberu et al. (2016) used 23 years of daily rainfall data to create a regression equation relating annual rainfall and the R-factor for the highlands of Malaysia. These simplified equations may be transferable to areas of similar climate that do not have the long-term detailed rainfall data required by the original R/USLE. The imperial units of erosivity are in hundreds of foot tonf inch acre$^{-1}$ hour$^{-1}$ year$^{-1}$, and multiplying by 17.02 will give the SI units of megajoule millimetre hectare$^{-1}$ hour$^{-1}$ year$^{-1}$ (Renard et al., 1997).

With the body of work that has been done in rainfall erosivity, some studies have managed to construct rainfall erosivity maps over large countries and regions. Panagos et al. (2017) have used pluviographic data from 63 countries to calculate rainfall erosivity and spatially interpolated the results to construct a global rainfall erosivity map at 30 arc-seconds resolution. Despite its coarse resolution, this global dataset can be used as a resource for rainfall erosivity in data-sparse regions. For the United States, Renard et al. (1997) details the procedure for obtaining rainfall erosivity values from their large national database. Renard et al. (1997) would be the recommended reference for study areas in the United States because of the extensive database that already exists for that country. For the European Union, Panagos et al. (2015d) constructed a rainfall erosivity map at 1km resolution and published descriptive statistics for R-values in each of the member countries. The interpolated map showed good agreement through cross-validation and to previous studies, but areas that had less rainfall stations and more diverse terrain caused higher prediction uncertainty (Panagos et al., 2015d). Using a large rainfall dataset,

da Silva (2004) constructed a spatially interpolated map of R-factors in Brazil whose trends showed agreement with previous work on rainfall erosivity in the country.

In areas that only have annual precipitation available, several equations and their studies can be used as a reference. In their global application, Naipal et al. (2015) published different R-factor equations depending on a study area's climate classification. One caveat is that the data for these equations had a large percentage of USA and European records, so resulting accuracy of R-factors might be better for those locations (Naipal et al., 2015). In tropical areas such as Southeast Asia, the R-factor by El-Swaify et al. (1987) as cited in Merritt et al. (2004) was used extensively in Thailand, the Philippines, and Sri Lanka. However, the units for the R-factor in this equation are given as tons hectare$^{-1}$ year$^{-1}$, which do not correspond to the original units used by R/USLE (Merritt et al., 2004). This lack of consistency regarding units is not uncommon in the reviewed literature, which sometimes fails to explicitly report the units used for the different factors. For example, Renard & Freimund (1994) report that the units of R-factor equations by Arnoldus (1977) were presumed to be in metric units. By being clear and consistent about units in R/USLE literature, future researchers can be more certain about the accuracy of their borrowed R-factor equations instead of presuming the units to be the same as the original R/USLE. Work by Bonilla & Vidal (2011) produced an R-factor equation for Chile and published erosivity values similar to those produced by work in areas of similar geography and geology. For New Zealand, Klik et al. (2015) proposed equations for calculating the annual R-factor and seasonal R-factor with coefficients that change depending on the study area's location within the country.

The usage of monthly precipitation data to determine the R-factor is due to monthly rainfall data being more readily available compared to detailed storm records (Renard & Freimund, 1994). Although annual rainfall estimates are sufficient, using monthly rainfall data to construct sub-annual R-factors and then aggregating those R-factors to an annual scale are useful in sites with large temporal variability in rainfall. Renard & Freimund (1994) used data from 155 stations with known R-factors based on the original USLE approach and related their R-factors to observed annual and monthly precipitation. These equations developed by Renard & Freimund (1994) in the west coast of USA were used in Ecuador (Ochoa-Cueva et al., 2015), and Honduras and El Salvador (Kim et al., 2005). Work by Arnoldus (1980) developed R-factor equations in West Africa that use monthly and annual precipitation. However, as described earlier these equations present a problem in terms of consistent units. In Southeast Asia, Shamsad et al. (2008) developed an R-factor equation in Malaysia that was used in the Philippines by Delgado & Canters (2012). In New Zealand, the monthly precipitation can be aggregated to seasonal precipitation and used in the equation for seasonal R-factor derived by Klik et al. (2015).

Monthly or better precipitation records are very useful in R/USLE applications because of the option of estimating soil loss at a monthly or seasonal scale, which can be useful in countries with high temporal variation of rainfall throughout the year. Monthly and seasonal erosion has been estimated by varying the R-factor depending on the monthly precipitation while leaving all the other factors constant (Ferreira & Panagopoulos, 2014; Kavian et al., 2011). Klik et al. (2015) emphasised the need to understand the drivers of soil erosion, including whether rainfall intensity had a stronger effect compared to mean annual rainfall. In an assessment of spatial and temporal variations in rainfall erosivity over New Zealand, December and January were associated with higher erosivities while August was associated with lowest erosivity (Klik et al., 2015). Similar

work by Diodato (2004) has cited the use of monthly erosivity data to be more useful with respect to managing crop growing cycles and tillage practices, especially during seasons where high rainfall erosivity is expected. In locations where there is a large temporal variation in rainfall throughout the year, the seasonal approach of estimating soil erosion is more important for sustainable land management (Ferreira & Panagopoulos, 2014)

To examine how different R-factor equations affected predicted soil erosion rates over the same study site, Benavidez (2018) tested three different equations over the ~157km$^2$ Mangatarere watershed in New Zealand. The equations by Klik et al. (2015) developed in New Zealand, along with the equations by Loureiro and Coutinho (2001) and Ferreira and Panagopolous (2014) developed in Portugal, were used to estimate annual and seasonal erosivity (Figure 1 and Table 1). All three equations consider and predict seasonal erosivity, and are from similar latitudes and developed in temperate to semi-arid environments.

For the same set of rainfall data, the three equations predicted different annual and seasonal values of erosivity. Regarding seasonal patterns of erosivity, Klik et al. (2015) predicted highest erosivity occurring during summer but lowest in winter and spring. This trend matches the national observations of the most erosive storms occur during summer, and the lowest occurring during winter (Klik et al., 2015). By contrast, both Loureiro & Coutinho (2001) and Ferreira & Panagopolous (2014) predicted highest erosivity during spring and lowest during summer. This variation is thought to be due to the Portugal equations

excluding days below 10.0mm of rainfall, which introduces a bias towards the erosive effects of short intense rainfall events while potentially excluding the erosive power of longer but less intense rainfall events. It is unsurprising that the New Zealand approach performed best in a New Zealand climate, but does demonstrate the risk of arbitrarily transferring equations between countries, even when geoclimatic conditions are not terribly dissimilar.

        These differences highlight the importance of understanding the regional applicability of rainfall erosivity equations.

In the reviewed R/USLE studies for this chapter, a common occurrence was using equations derived in different countries and regions without much justification why those equations were chosen with little consideration for their suitability. These studies also did not publish any testing of how different R-factors produce different erosivity values from the same input dataset. The purpose of testing the different R-factors is to illustrate this variation and encourages future users of R/USLE to do the same sensitivity testing in their area.

In summary, there are many rainfall erosivity datasets and equations in the R/USLE literature that can be used by new researchers applying the RUSLE to their study area. The erosivity dataset produced by Panagos et al. (2017) is recommended for areas with no rainfall data or in ungauged catchments since this is a raster dataset with a global coverage (~30 arcsecond resolution) and is freely available. For areas in the European Union, work by Panagos et al. (2015d) has produced a rainfall erosivity map with regional coverage at ~1km resolution. These datasets can also be used to validate the erosivity factors

calculated at the national or catchment scale. If annual precipitation and the study area's Köppen-Geiger classification are known, Naipal et al. (2015) has published rainfall erosivity equations and values for 17 different climate zones. Several studies have published erosivity equations for tropical areas: da Silva (2004) for Brazil, Shamshad et al. (2008) for Malaysia, and Jain & Das (2010) for India. For arid areas, Arnoldus (1980) as cited in Renard & Freimund (1994) has derived erosivity equations for Morocco and other locations in West Africa. Many other equations are found in Table 2 and choosing several for sensitivity

testing is recommended for future R/USLE applications. It is also important to test against observed data or R-factors derived by previous applications in the same study area or in study areas with similar climatic regimes.

## 2.2 Soil erodibility factor (K)

The K-factor represents the influence of different soil properties on the slope's susceptibility to erosion (Renard et al., 1997). It is defined as the "mean annual soil loss per unit of rainfall erosivity for a standard condition of bare soil, recently tilled up-and-down slope with no conservation practice" (Morgan, 2005). The K-factor essentially represents the soil loss that would occur on the R/USLE unit plot, which is a plot that is 22.1m long, 1.83m wide, and has a slope of 9% (Lopez-Vicente et al., 2008).

Higher K-factor values indicate the soil's higher susceptibility to soil erosion (Adornado et al., 2009). In the R/USLE, Wischmeier and Smith (1978) and Renard et al. (1997) use an equation that relates textural information, organic matter, information about the soil structure and profile-permeability with the K-factor or soil erodibility factor. However, other soil classifications might not include soil structure and profile-permeability information that matches the information required by R/USLE nomograph. Hence, alternative equations have been developed that exclude the soil structure and profile-permeability (Table 3). The question of which equation to use depends on the availability of soil data. Where only the textural class and organic matter content is known, Stewart et al. (1975) have approximated K-factor values based on these inputs. Similar to the R-factor, the imperial units of soil erodibility are in ton acre hour hundreds of acre$^{-1}$ foot$^{-1}$ tonf$^{-1}$ inch$^{-1}$, and multiplying by 0.1317 gives the erodibility in SI units of metric ton hectare hour hectare$^{-1}$ megajoule$^{-1}$ millimetre$^{-1}$ (Renard et al., 1997).

Although seemingly relatively straightforward, the K-factor equation proposed by Wischmeier and Smith (1978) comes with a few limitations regarding soil type. This equation was developed using data from medium-textured surface soils in the Midwestern USA, with an upper silt fraction limit of 70% (Renard et al., 1997). An equation for volcanic soils in Hawaii was proposed by El-Swaify & Dangler (1976) as cited in Renard et al. (1997), but is only appropriate for soils similar to Hawaiian soils and not for all tropical soils. Despite these limitations, many studies outside the USA have used the original Wischmeier & Smith (1978) K-factor equation (Table 3). Being aware of the regional specificity of K-factor equations is important, and using different K-factor equations in one study area to find a range of soil erodibility could be a way of testing their applicability.

Similar to the sensitivity analysis of the R-factor equations, testing different K-factor equations to see the variation in erodibility values, and then comparing these K-factors with published values from similar soils would be a good way to test applicability. For the spatial coverage of European Union, a soil erodibility raster dataset (~500m resolution) is available for validation (Panagos et al., 2014). David (1988) and Dymond (2010) have published K-factor values for soils of different textural classes (e.g. clay, loam, etc.) that can be used if only soil texture is known (Table 4 and Table 5). However, the values published by Dymond (2010) are broad and do not account for soils with mixed texture, while the values of David (1988) are based on soils in the Philippines. Like the R-factor, it is important to check the derived K-factor values for the site-specific soil against previously published K-factor values for comparable sites and soil types.

**2.3 Slope length (L) and steepness (S) factor**

The LS-factor represents the effect of the slope's length and steepness on sheet, rill, and inter-rill erosion by water, and is the ratio of expected soil loss from a field slope relative to the original USLE unit plot (Wischmeier & Smith, 1978). The USLE method of calculating the slope length and steepness factor was originally applied at the unit plot and field scale, and the RUSLE extended this to the one-dimensional hillslope scale, with different equations depending on whether the slope had a gradient of more than 9% (Renard et al., 1997; Wischmeier & Smith, 1978). Further research extends the LS-factor to topographically complex units using a method that incorporates contributing area and flow accumulation (Desmet & Govers, 1996). The USLE and RUSLE method of calculating the LS-factor uses slope length, angle, and a parameter that depends on the steepness of the slope in percent (Wischmeier & Smith, 1978).

One of the criticisms of the original USLE method of calculating LS-factor is its limited applicability to complex topography. With advances in GIS technology, the method of determining the LS-factor as a function of upslope contributing area or flow accumulation and slope has risen in popularity (Table 6). The use of digital elevation models (DEMs) to calculate the upslope contributing area and the resulting LS-factor allows researchers to account for more topographically complex landscapes (Moore & Burch, 1986; Desmet & Govers, 1996). Desmet and Govers (1996) have also built on this method through showing its application in a GIS environment over topographically complex terrain when compared to the original method proposed by Wischmeier and Smith (1978). This method of using flow accumulation for slope length and steepness explicitly accounts for convergence and divergence of flow, which is important when considering soil erosion over a complex landscape (Wilson & Gallant, 2000). It is possible to use this method to calculate the LS-factor over a large extent, but a high-resolution DEM is needed for accurate representation of the topography. The resolution required depends on the study area's scale. The relatively coarse globally available DEMs (~30m at best) are less suited to field and sub-catchment scale studies where it may be important to capture effects of micro-topography.

The original equations for LS-factor assume that slopes have uniform gradients and any irregular slopes would have to be divided into smaller segments of uniform gradients for the equations to be more accurate (Wischmeier & Smith, 1978). At the plot or small field scale, this manual measurement of slopes and dividing into segments may be manageable, but less useful at larger scales. In terms of practicality, Desmet & Govers (1996) have reported studies of this method applied at a watershed scale with the disadvantages of it being time-consuming. Studies in Iran and the Philippines have implemented the R/USLE methods within a GIS environment by calculating the LS-factor for each raster cell in a DEM, essentially treating each pixel as its own segment of uniform slope (Bagherzadeh, 2014; Schmitt, 2009).

As explained above, the method of using flow accumulation, upslope contributing area, and slope in a GIS environment has gained popularity due to its ability to explicitly account for convergence and divergence of flow, thus capturing more complex topography (Wilson & Gallant, 2000). The flow accumulation method was applied at the scales of watersheds and regions (as shown in Table 6) and has even been applied by Panagos et al. (2015a) at the scale of the European Union using a 25m DEM. The only thing limiting users is the availability of high-resolution DEMs and the trade-off between processing time

and accuracy. The original R/USLE methods require only slope angle and length, operate over a single cell in a DEM by treating it as a uniform slope, and take less processing time compared to the method using flow accumulation. However, the user must remember that this cannot capture the convergence and divergence of flow and thus sacrifices accuracy for time.

Additionally, the issue of limited vertical accuracy in global and many national DEMs confounds the uncertainties associated with coarse cell sizes. Further work on understanding the appropriate horizontal resolution and vertical accuracy of DEMs used for soil erosion predictions at the sub-catchment or field scales is suggested.

Benavidez (2018) investigated use of high-resolution DEMs (15m and finer), finding the methods that only used slope length and steepness were adequate at delineating large vulnerable areas at the watershed scale. However, the methods using flow accumulation performed significantly better at the sub-watershed or field scale (Benavidez, 2018).

In summary, the choice of which LS-factor method to use is dependent on the spatial resolution of the DEM, availability of computing resources, and the scale of the study site. Since DEMs with resolution coarser than ~100m do not accurately capture the flow network of a catchment (Panagos et al., 2015a), sites with coarse DEMs should use the LS-factor methods that account for only slope length and steepness instead of using more computing resources to use methods that account for flow accumulation. At the national, regional, or watershed scale, delineating large areas vulnerable to soil loss is more useful due to the ease of managing these areas at such large scales, and the methods that use only slope length and steepness are recommended. For sub-watershed or field studies and with sufficiently fine DEMs (~15 or finer), using LS-factor methods that account for flow accumulation are more useful for identifying the most critical areas of vulnerability for targeted management approaches.

## 2.4 Cover and management factor (C)

The cover and management factor (C) is defined as the ratio of soil loss from a field with a particular cover and management compared to a field under "clean-tilled continuous fallow" (Wischmeier & Smith, 1978). The R/USLE uses a combination of sub-factors such as impacts of previous management, canopy cover, surface cover and roughness, and soil moisture on potential erosion to produce a value for soil loss ratio, which is used with R-factor to produce a value for C-factor (Renard et al., 1997). This method requires extensive knowledge of the study area's cover characteristics including agricultural management and may be suitable at field or farm scale, but monitoring all these characteristics at the watershed scale may not be feasible.

A simpler method of determining the C-factor is referencing studies that have reported values for similar land cover, or from studies done in the same area or region. Table 8 and Table 9 give a broad overview of C-factors for different cover types and common crops. Wischmeier & Smith (1987) also include the effect of percent ground cover, reporting C-factor values for the same cover type over a range of cover percentage and condition. Morgan (2005) and David (1988) have reported values for the different growth stages of the same types of trees. A simple method of creating a C-factor layer us by using lookup tables to assign C-factor values to the land cover classes present in the study area. When using C-factors from literature, it is important to note the definition of land cover type between two countries may vary. For example, land classified as forest in

one country may be different in terms of vegetation cover or type compared to forest in another country (e.g. differences in pine forests and tropical forests). Therefore, it is crucial to understand the differences between land cover classifications before applying C-factor values from literature. Van der Knijff et al. (2000) cites the large spatial and temporal variations in cover and crop over a large region such as the European Union as another reason why using the lookup table-based approach is inadequate and tedious.

To address this, another method of determining the C-factor is through the Normalized Difference Vegetation Index (NDVI) estimated from satellite imagery. Although there are NDVI layers available, these are limited by geographical coverage, date of acquisition, and resolution. The MODIS NDVI dataset made by Caroll et al. (2004) at 250m resolution covers the USA and South America[2]. NASA produced a global dataset of NDVI values at 1-degree resolution for the timespan of July 1983 to June 1984, making it suitable for studying historical soil erosion but not necessarily for the current state of land cover[3].

In areas where ready-made NDVI products are unavailable, authors have used satellite imagery to obtain NDVI such as AVHRR or Landsat ETM (Van der Knijff et al., 2000; De Asis & Omosa, 2007; Ma et al., 2001 as cited in Li et al., 2014). De Asis & Omasa (2007) related C-factor and NDVI through fieldwork and image classification; determining C-factor at several points within the study area using the R/USLE approach and relating it to the NDVI through regression correlation analysis. For larger study areas, this may not be feasible such as in the European Union where Van der Knijff et al. (2000) determined NDVI from satellite imagery and created an equation based on its positive correlation with green vegetation (Table 7). This approach enabled them to create a C-factor map over the European Union. However, C-factors were unrealistically high in some areas such as woodland and grassland, so values for those areas were taken from literature.

An advantage of using is NDVI that researchers can determine sub-annual C-factors if there is satellite imagery available, which can lead to understanding the contribution of cover to seasonal soil erosion and identifying critical periods within the year were soil erosion is a risk (Ferreira and Panagopoulos, 2014). Similar methods have been applied in Brazil by Durigon et al. (2014), Greece by Alexandridis et al. (2015), and Kyrgyzstan by Kulikov et al. (2016). Determining C-factors at the seasonal scale is important because vegetation cover can change throughout the year due to agricultural and forestry practices. In study areas with a high temporal variation of rainfall throughout the year, seasonal vegetation can play a big part in exacerbating or mitigating soil erosion.

To summarise, the choice of which method to use depends on the scale of the study area, reported C-factors for similar cover, and availability of high-resolution imagery. For small-scale studies, it is more feasible to determine the C-factors through fieldwork. If previous R/USLE studies have reported C-factors for cover similar to the study area, those values can be used for the table-based approach. Lastly, high-resolution imagery can be used to determine the study area's NDVI. At small scales and with a good understanding of differences in land cover classifications, pulling values from literature may be the most efficient choice but at larger regional scales, this may become tedious. At larger scales, high-resolution satellite imagery

[2] http://glcf.umd.edu/data/ndvi/
[3] https://data.giss.nasa.gov/landuse/ndvi.html

may be available to determine NDVI but authors must be mindful of its acquisition date in relation to their study period, and data quality and image processing issues such as dealing with cloud cover and creating aggregating images from multiple satellite passes (Van der Knijff et al., 2000; Kulikov et al., 2016).

## 2.5 Support practice factor (P)

The support practice factor (P) is defined as the ratio of soil loss under a specific soil conservation practice (e.g. contouring, terracing) compared to a field with upslope and downslope tillage (Renard et al., 1997). The P-factor accounts for management practices that affect soil erosion through modifying the flow pattern, such as contouring, strip-cropping, or terracing (Renard et al., 1997). The more effective the conservation practice is at mitigating soil erosion, the lower the P-factor (Bagherzadeh, 2014). Like the C-factor, values for P-factors can be taken from literature and if there are no support practices observed, the P-factor is 1.0 (Adornado et al., 2009). The P-factor can also be estimated using subfactors, but the difficulty of accurately mapping support practice factors or not observing support practices leads to many studies ignoring it by giving their P-factor a value of 1.0 as seen in Appendix 1 (Adornado et al., 2009; Renard et al., 1997; Schmitt, 2009).

Another possible reason why studies may ignore P-factor is due to the nature of their chosen C-factors. Some C-factors already account for the presence of a support factor such as intercropping or contouring. For example, Morgan (2005) and David (1988) give C-factors for one type of crop, but with different types of management (Table 10). Despite the P-factor being commonly ignored, a number of studies have reported possible P-factors for different kinds of tillage, terracing, contouring, and strip-cropping (Table 11). The P-factor has a significant impact on the estimation of soil loss. For example, a P-factor of 0.25 for zoned tillage reflects the potential for this management factor to reduce soil by 75% loss compared to conventional tillage (P-factor: 1.00). At suitably detailed scales and with enough knowledge of farming practices, using these P-factors may lead to a more accurate estimation of soil loss. Additionally, these P-factors can be used in scenario analysis to understand how changing farming practices may mitigate or exacerbate soil loss. An application of R/USLE in the Cagayan de Oro catchment in the Philippines showed, through scenario analysis, that soil conservation practices such as agroforestry and alley-cropping could potentially lead to large decreases in soil loss compared to the baseline scenario (Benavidez, 2018).

In summary, including the P-factor in R/USLE applications is important because of the significant effects that some management practices can have on reducing soil loss compared to conventional tillage. The P-factor is useful for studies where different management practices are being considered for the same site as it can elucidate which practices are more beneficial for soil conservation.

## 3 Limitations of R/USLE

This section presents a few of the key limitations of the R/USLE: regional applicability, uncertainties associated with the model, input data and validation, and representing other types of erosion.

The most commonly cited limitation of the R/USLE models is their reduced applicability to regions outside of the United States of America (Aksoy & Kavvas, 2005; Naipal et al., 2015; Sadeghi et al., 2014). The original USLE was formulated based on soil erosion studies on agricultural land in the USA. When applied to different climate regimes and land cover conditions, this may lead to greater uncertainties associated with estimates of average annual soil loss (Kinnell, 2010). Since

the R/USLE parameters were developed based on small sacale studies of agricultural plots, there are also uncertainties associated with upscaling the original USLE to the catchment or regional scale (Nagle et al., 1999; Naipal et al., 2015). Wischmeier & Smith (1987) have also warned that using the R/USLE in conditions extremely different from the agricultural conditions the model was formulated under may lead to extrapolation error. Of the studies reviewed for this paper (Table A1), most applications were done on catchments with predominantly agricultural land use, but under a range of different climatic

conditions.

Sensitivity analysis and testing which R/USLE sub-factors suit particular study sites is one method of addressing the R/USLE's regional applicability. Like the Mangatarere application method in Section 2.1, other studies have tested multiple R-factor equations on the same dataset to determine which equation was most appropriate for their study site (Eiumnoh, 2000; Benavidez, 2018). Their derived R-factor values were compared to the values for catchments with similar climate and rainfall,

or to maps of R-factor at larger spatial scales (Panagos et al., 2017). To reduce uncertainty in accounting for land use, work by Post & Hartcher (2005) recommended using C-factor values for specific land cover classifications (e.g. specific crops, forest growth stages) instead of values for broad land cover categories (e.g. agriculture, forest). Although C-factor values can be taken from literature or determined in-situ, an extensive literature review compiling potential soil loss rates of different crop and forest covers compared to likely soil loss rates of bare soil can be used to determine likely C-factor values of a particular

site. Improvements and modifications to the R/USLE sub-factors have made it applicable to larger spatial scales, including a coarse resolution representation at the global scale (Naipal et al., 2015). The pan-European application by Panagos et al. (2015a) showed setting a maximum value for slope steepness of 50% (26.6 degrees) would prevent significantly large LS-factor values and account for the absence of soil on such steep slopes. Assembling published estimates of R/USLE sub-factors from different climatic regions and soil types would help in sensitivity testing R/USLE equations, deciding the most

appropriate equation to use, and verifying the derived R/USLE sub-factor values.

The uncertainties associated with the R/USLE, and arguably soil erosion modelling in general, stem from several factors: the inability of models to capture the complex interactions involved in soil loss, the low availability of long-term reliable data for modelling, and the lack of soil erosion observational data for model validation, especially in data-scarce environments. The simplicity of the R/USLE allows usage in locations where there is insufficient data for more complex

models that require large input datasets (de Vente & Poesen, 2005; Hernandez et al., 2012). Of the studies reviewed, very few critically discuss the uncertainties associated with the R/USLE but those that do offer several ways to overcome these uncertainties.

Since the R/USLE does not account for all the complex interactions associated with soil erosion, and its predicted soil erosion rates should be taken as best estimates rather than absolute values (Wischmeier & Smith, 1987). Some applications

have chosen to display their soil loss results as categorical to produce maps that show low, medium, or high areas of vulnerability instead of showing annual average amounts (Adornado et al., 2009; Schmitt, 2009). The R/USLE is a good first attempt at identifying vulnerable areas and estimating soil loss for a landscape at the baseline scenario due to the model's relative simplicity and few data requirements (Aksoy & Kavvas, 2005). The R/USLE is also useful for doing scenario analysis

to check whether changing land use or management practices would either exacerbate or mitigate soil loss, making it useful for comparison purposes (Merritt et al., 2004; Nigel & Rughooputh, 2012).

      Validating the soil erosion rates produced by the R/USLE is difficult because of the lack of easily obtainable observational soil erosion records, especially in data-scarce environments. Out of the R/USLE applications reviewed for this paper, ~30% presented explicit comparisons between their modelled soil loss from R/USLE and observed soil loss, modelled

soil loss from R/USLE and other models (1 study), and soil loss from multiple models and observed soil loss (1 study).

      One study compared the soil loss rates predicted by the RUSLE to estimates of the physically-based WEPP (Water Erosion Prediction Project) model. Amore et al. (2009) compared RUSLE and WEPP and found that the modelled to observed ratio of soil loss of WEPP (0.7) was better than RUSLE (0.2) for the Trinità basin. However, both RUSLE and WEPP over-predicted sediment yield by up to five times the observed value for the nearby Ragoleto basin (Amore et al., 2009). Although

WEPP also estimates rill and inter-rill erosion, WEPP is a continuous daily model that accounts for deposition and sediment delivery that RUSLE does not predict (Aksoy & Kavvas, 2005).

      Another study compared the soil loss estimates of the RUSLE to USPED to each other, and to observed data. In a comparison between the RUSLE and USPED, the modelled to observed ratio of soil loss was almost unity for the USPED but 0.86 for the RUSLE (Aiello et al., 2014). The USPED model builds and improves on the RUSLE sub-factors through its ability

to incorporate overland flow and sediment transport through the landscape (Aiello et al., 2014; Zakerinejad & Maerker, 2015).

      Based on the remaining studies that reported comparisons of modelled RUSLE soil loss to observed soil loss, the ratio of modelled to observed ranged from extreme under-prediction at 0.04 to over-prediction at over three times the observed values. The applications where RUSLE severely under-predicted soil loss cited the model's inability to account for gully erosion and mass wasting as one of the reasons for estimation errors, thus underscoring the importance of including these types

of erosion in future improvements to RUSLE (Dabney et al., 2012; Gaubi et al., 2017). Another issue is differences in temporal and/or spatial resolution and sometimes differing time scales between modelled and observed estimates. Average observations based on occasional grab samples of sediment in streams may not well represent the monthly to annual sediment loads the R/USLE is attempting to estimate. In another example, López-Vicente et al. (2008) compared observed to modelled values and had a modelled to observed soil loss ratio of 0.62. However, the "observed" soil loss was based on [137]Cs measurements

that were indicative of average soil loss values for the past forty years while the model values were based on 1997 to 2006 driving data. During this period, the study area experienced lower precipitation and thus had lower modelled soil loss measurements compared to the soil loss derived from the [137]Cs records (López-Vicente et al., 2008).

      As stated earlier, the regional applicability of the RUSLE is a limitation that requires the sub-factors to be adjusted and modified based on the specific characteristics of the researcher's study site. Nakil & Khire (2016) and Abu Hammad et al.

(2005) show this important practice in RUSLE applications in their studies. Through testing and refining their method of accounting for topography through the LS-factor, the ratio of modelled to observed soil loss ranged from 0.8 to almost unity (Nakil & Khire, 2016). The initial application of RUSLE of Abu Hammad et al. (2005) over-estimated soil loss by a factor of three but with adjustments to the sub-factors based on local data on soil moisture, land cover, and support practices, the model

error was reduced to 14%. The importance of adjusting RUSLE with the availability of more detailed data was further shown in the pan-European study of Panagos et al. (2015e) where detailed soil, topography, land cover, and management practices allowed the researchers to refine their application where most of the modelled to observed soil loss ratios were very good (0.9 to 1.3). In the validation areas where the soil loss comparisons were not good, further local testing and refining of the RUSLE sub-factors is seen as an area to improve the model results (Beskow et al., 2009; Ozsoy et al., 2012; Panagos et al., 2015e).

A global soil erosion study using RUSLE has been accomplished by Borrelli et al. (2017) using the rainfall erosivity map generated by Panagos et al. (2017) that showed comparable results to regional and local soil erosion estimates, and good agreement with global soil erosion datasets such as the Global Assessment of Human-induced Soil Degradation (GLASOD) dataset[4].

    Future work in soil erosion literature could include assembling a comprehensive database of global, regional, and

national soil erosion rates to allow comparison between soil erosion modelling methods, not just R/USLE results. A proxy for understanding soil erosion is water quality data such as total suspended solids (TSS) that includes sediment delivery and organic sources (Schmitt, 2009; Russo, 2015). However, TSS usually excludes the larger and heavier bedload sediments that could be resulting from mass wasting events or erosion (Nagle et al., 1999). Nevertheless, water quality data is useful for inferring likely temporal patterns of soil erosion or the sediment yield after during seasons of heavy rainfall or after extreme

events. Ground-truthing or analysis of satellite imagery is another useful method of validating the R/USLE results, as the areas of extreme erosion risk can be checked for physical evidence of soil loss occurrence (De Asis & Omasa, 2007; Adornado & Yoshida, 2010; Nontananandh & Changnoi, 2012). The soil loss estimates can be validated against observations from similar catchments, recorded events of mass wasting, or against larger scale soil loss studies at the national or regional scale (Životić et al., 2012; Panagos et al., 2015e; Nakil & Khire, 2016).

Lastly, a frequently cited limitation is that the R/USLE estimates soil loss through sheet and rill erosion, but not from other types of erosion such as gully erosion, channel erosion, bank erosion, or from mass wasting events such as landslides (Nagle et al., 1999; Wischmeier & Smith, 1978). By excluding these types of erosion, the R/USLE may underestimate the actual soil loss (Thorne et al., 1985). The model also does not account for deposition, leading to overestimation, or sediment routing (Desmet & Govers, 1996; Wischmeier & Smith, 1978). Since it does not predict the sediment pathways from hillslopes

to water bodies, it is difficult to analyse possible effects on downstream areas, such as pollution or sedimentation (Jahun et al., 2015). One of the possible methods to link the R/USLE results to sediment delivery to streams is using the stream delivery ratio (SDR) defined as "the ratio of the sediment delivered at a location in the stream system to the gross erosion from the

---

[4] https://www.isric.online/projects/global-assessment-human-induced-soil-degradation-glasod

drainage area above that point" (Yoon et al., 2009). This parameter varies depending on the gradient, slope shape, and length and can also be influenced by land cover, roughness, etc. (Wu et al., 2005). Given that it is influenced by similar characteristics as the R/USLE, future work can include combining the R/USLE with the SDR to estimate sediment delivery to streams, but also avoiding possible double-counting. These two limitations of deposition and routing are linked to the model's representation of more topographically complex terrain, and previous studies have attempted to address it by improving on the LS-factor by incorporating upstream contributing area (Desmet & Govers, 1996; Moore et al., 1991). A more detailed discussion of addressing these limitations is in Section 4.1.

Despite these drawbacks, the USLE family of models is still widely used because of is relative simplicity and low data requirements compared to more complex physically based models. Studies around the world continue to improve R/USLE parameterisation and application in different climate regimes and locations.

## 4 Future directions

Since the R/USLE and its family of models are used over different geographic locations and climate types, it is important for future research to build on them and improve their representation of real-world soil loss. Some of the future directions include incorporating soil loss from other types of erosion, estimating soil loss at seasonal or sub-annual temporal scales, and improving the consistency of formulae and units in the scientific literature.

### 4.1 Representing other types of erosion

As previously discussed in the limitations section, the R/USLE does not account for all erosion types. This section mostly discusses possible extensions to include gully erosion, but further work to incorporate channel/bank erosion and mass wasting events must also be done.

The inability of R/USLE to account for soil losses due to ephemeral gullies can lead to under-prediction of soil loss estimates (Thorne et al., 1985). These ephemeral gullies are small channels that form due to the erosive action of overland flow during a rainfall event (Momm et al., 2012). Gully erosion can contribute a significant amount of sediment loss, for example gully erosion is estimated to contribute between 30% to 50% of soil loss from a range of catchments in New Zealand (Basher et al., 2013). Desmet & Govers (1996) recommended that delineation of ephemeral gullies, such as through the Compound Topographic Index (CTI) developed by Thorne et al. (1985), combined with R/USLE could improve the identification of vulnerable areas within a watershed. The CTI of Thorne et al. (1985) uses topographic analysis to predict locations and soil loss rates of ephemeral gullies based on upstream drainage area, slope, and the planform curvature. Hence, the combination of CTI and the R/USLE is a promising direction for including gully erosion but care must be taken in coupling these models because both already account for upstream drainage area and slope. Simply adding their soil loss rates could lead to "double-counting" and requires further research to determine the threshold values of CTI and LS-factor over which ephemeral gullying is likely (Benavidez, 2018).

Work along these lines, combining the effect of rill and sheet erosion with gully erosion, was done by Momm et al. (2012) in Kansas, and by Zakerinejad and Maeker (2015) in the Mazayjan watershed in Iran. Momm et al. (2012) combined several types of erosion: sheet and rill, gully, and bed and bank erosion, with the sheet and rill erosion estimated using the R/USLE model. They used varying critical CTI thresholds to iteratively generate potential locations of ephemeral gullies,

identify sub-watersheds prone to gully erosion, and then used scenario analysis to estimate reductions in sediment yields under conservation practices (Momm et al., 2012). One of the limitations of the Momm et al. (2012) study was that they only had a coarse resolution DEM. Since ephemeral gullies are small features (typically a few metres wide and ~25cm deep), higher-resolution DEMs such as those derived from LiDaR data would be better for analysis of these topographic features. The Unit Stream Power Erosion Deposition Model (USPED), which is similar to the R/USLE model, has also been used to estimate rill

and sheet erosion rates with a stream power index (SPI) approach to estimate gully erosion rates (Zakerinejad & Maerker, 2015). Zakerinejad & Maerker (2015) estimated gully erosion in tons hectare$^{-1}$ year$^{-1}$ and combined it with the estimates from the USPED model to produce a map showing potential erosion and deposition within their study area. Hence, there are precedents as well as a need to combine erosion estimates from R/USLE with a procedure that accounts for gully erosion for more effective land management.

**4.2 Seasonal erosion vulnerability**

R/USLE applications usually estimate soil loss at the annual timescale, and the MUSLE estimates soil loss from a single storm event (Renard et al., 1997; Sadeghi et al., 2014). As seen in the review of methods to calculate rainfall erosivity, many different studies have attempted to estimate the R-factor, underscoring its importance to soil erosion research. However, estimating the R-factor at the annual timescale does not account for seasonal variations in rainfall. It is useful for land

management to understand seasonal variations in soil erosion vulnerability because of the dual effect of rainfall and land cover on soil loss, and the effect of rainfall on land cover (Kulikov et al., 2016). For example, when a season of heavy rainfall coincides with low vegetation cover, the risk of soil erosion increases considerably (Ferreira & Panagopoulos, 2014). Thus, most of the studies around seasonal estimations of soil loss revolve around changes in land cover and rainfall. The soil erodibility (K-factor) can vary too due to changes in permeability and the effects of freezing and thawing, but it is less

frequently studied compared to variations in land cover and rainfall (López-Vicente et al., 2008).

Studies that incorporate seasonality in the R/USLE commonly compute R-factors and C-factors at monthly or seasonal time scales. Lu & Yu (2002) computed monthly R-factors in Australia, which was then used in a later study that computed C-factors based on satellite imagery and the NDVI, to produce monthly maps of soil erosion vulnerability over the entire Australian continent (Lu et al., 2003; Lu & Yu, 2002). The method of estimating C-factors using NDVI is popular due to the

30 availability of remotely-sensed imagery, and the capability of processing datasets with relative expedience compared to time-consuming fieldwork. Other studies have used the NDVI and similar characteristics to estimate monthly and seasonal C-factors in Brazil, Greece, and Kyrgyzstan (Alexandridis et al., 2015; Durigon et al., 2014; Ferreira & Panagopoulos, 2014; Kulikov et

al., 2016; Panagos et al., 2012). The C-factors can also be estimated monthly through the method recommended by R/USLE, but requires knowledge of prior land use, canopy cover, surface roughness, and soil moisture (López-Vicente et al., 2008).

Monthly or seasonal estimations of rainfall factors are more useful to land management planning around crop growth cycles and tillage practices (Diodato, 2004). Studies have used different methods to calculate R-factors, with data requirements ranging from per-storm basis to annual averages. To estimate monthly and seasonal estimations, the required rainfall data can be as fine as individual storm intensity to use the R/USLE method, or be as coarse as average monthly rainfall. Diodato (2004) in Italy and Kavian et al. (2011) used the R/USLE method to calculate storm energy and summed these up per month and season to obtain R-factors. Other studies used daily and monthly rainfall to calculate monthly R-factors and combine them for seasonal R-factors (Alexandridis et al., 2015; Kavian et al., 2011; López-Vicente et al., 2008; Lu et al., 2003; Panagos et al., 2015d; Shamshad et al., 2008). The results of these studies focused on identifying high and low periods of the landscape's vulnerability to soil erosion, depending on combinations of rainfall intensity and land cover.

At the baseline scenario, applying the R/USLE can give management an idea of which areas are vulnerable to soil erosion. Previous work by Alexandridis et al. (2015) and Ferreira & Panagopoulos (2014) have looked at seasonal variations in soil loss due to land cover using satellite imagery from different times of the year. These approaches are useful in determining soil loss based on previous or existing land cover, but the next step is using scenario analysis to help land management. Scenario analysis can include a myriad of options: expanded urban areas or development, changing crop rotation cycles, or applying support practices in steep or upland areas. By adding seasonal effects, it gives additional knowledge of when these vulnerable areas may be even more vulnerable. Thus, by using scenario analysis, management can test different types of crop and support practices to see their possible effect on soil erosion mitigation. Soil erosion also affects water quality because of sediment delivery to streams and rivers, which raises concerns about access to clean water for drinking and for recreational use. Therefore, understanding seasonal soil erosion is beneficial to local government who can address potential sources of sediment delivery before the problem occurs and be more proactive in their land management.

To summarise, modelling the sub-annual variations of soil erosion and sediment yield is important for understanding how temporal variations in rainfall affect soil loss, changes in the spatial distribution of erosion-prone areas over crop growth and tillage cycles, and potential seasonal changes in water quality due to changes in seasonable distributions of heavy rainfall or other extreme events. Seasons with higher heavy rainfall will have a higher possibility of soil loss and mass wasting events, which in turn have a degrading effect on water quality and can cause destruction of infrastructure, putting communities and lives in danger. Over the crop growth and tillage cycles, the potential sediment yields to streams will change and this has implications for farmers and land management who must abide by water quality standards. Modelling at the annual timescale is insufficient to capture these seasonal or monthly changes in potential soil loss, which are more important to land management planning, and thus underscores the utility of doing modelling at the sub-annual scale.

**4.3 Consistency in units**

The USLE was originally developed using imperial units and although the handbook provides conversion factors to convert to metric, there are still issues within the scientific literature regarding units. In the process of this review, it was noted that although most studies used the metric units for R-factor and K-factor, there were other studies that did not report their units or had units that were not the imperial or metric units of R/USLE. Since the original R/USLE was formulated with US customary units, researchers must be careful to use the correct units and conversions to metric (Renard & Freimund, 1994). To convert from imperial to metric units, Renard et al. (1997) recommends a conversion factor of 17.02 for R-factor and 0.1317 for K-factor. As mentioned in Section 3, there are uncertainties associated with the R/USLE and publishing sub-factor values and soil loss estimates for future reference by other researchers is a potential way of reducing some of those uncertainties. The problem of unclear or inconsistent units causes problems for future researchers in terms of adapting the rainfall erosivity or soil erodibility equations for their own study sites, underscoring the need for clear and explicit reporting of units in the R/USLE literature.

**Summary and conclusion**

At first glance, the USLE and its family of models seems like a relatively straightforward linear model. However, this review shows the difficulty in finding the most appropriate method of calculating its sub-factors depending on location, availability of data, and previous studies done in nearby or similar regions. This paper reviewed the different components of the Universal Soil Loss Equation (USLE) and its updated form, the Revised Universal Soil Loss Equation (RUSLE). Different studies around the world were collected and analysed to compile how they adapted R/USLE to their unique conditions, how they had estimated the R/USLE sub-factors with limited data availability, and how these methods have been used by subsequent soil erosion studies. At the end of each sub-factor section, a brief summary is given outlining which datasets and equations would be useful for new users depending on their location and data availability. Each sub-factor section clarifies some of the assumptions and limitations associated with the original R/USLE models, and how users can overcome some of the uncertainties associated with these sub-factors. One common theme in the sub-factor reviews is that sensitivity testing of the sub-factors should be done by future R/USLE applications by trialling several equations for one sub-factor before using it in the final soil erosion estimates.

This paper also presented the limitations of the R/USLE, mainly the uncertainties associated with the simple empirical model, uncertainties with data availability and validation, and the model's inability to account for types of soil erosion other than rill or inter-rill erosion. Lastly, the paper outlined some key few future directions for R/USLE research: incorporating soil loss from other types of soil erosion, importance of estimating soil loss at sub-annual scales and recommended equations, and consistency in reporting units in future literature. To represent gully erosion, the Compound Topographic Index (CTI) was briefly discussed while the Sediment Delivery Ratio (SDR) was also presented to account for linking soil loss to sediment delivery to streams. The importance of doing sub-annual soil loss or seasonal erosion modelling is important due to some study

areas having high temporal variation of rainfall throughout the year, or having varying crop growth and tillage cycles, both being factors that can impact potential soil loss. Greater transparency in reporting the sub-factor units, sub-factor values, and soil loss estimates is important to decrease uncertainty when future R/USLE applications borrow sub-factor equations and values from previous studies. The limitations section addresses the fourth objective of this review.

In the end, the choices made regarding applications of the R/USLE depend on the kind of data that is available for a study area, and how they can adapt or change information from other studies to suit their area's particular climate, soil type, topography, typical land cover, and support practices.

**Acknowledgements**

Much of the content of this paper was supported by the Victoria University of Wellington Doctoral Scholarship as part of the
PhD research of R. Benavidez.

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

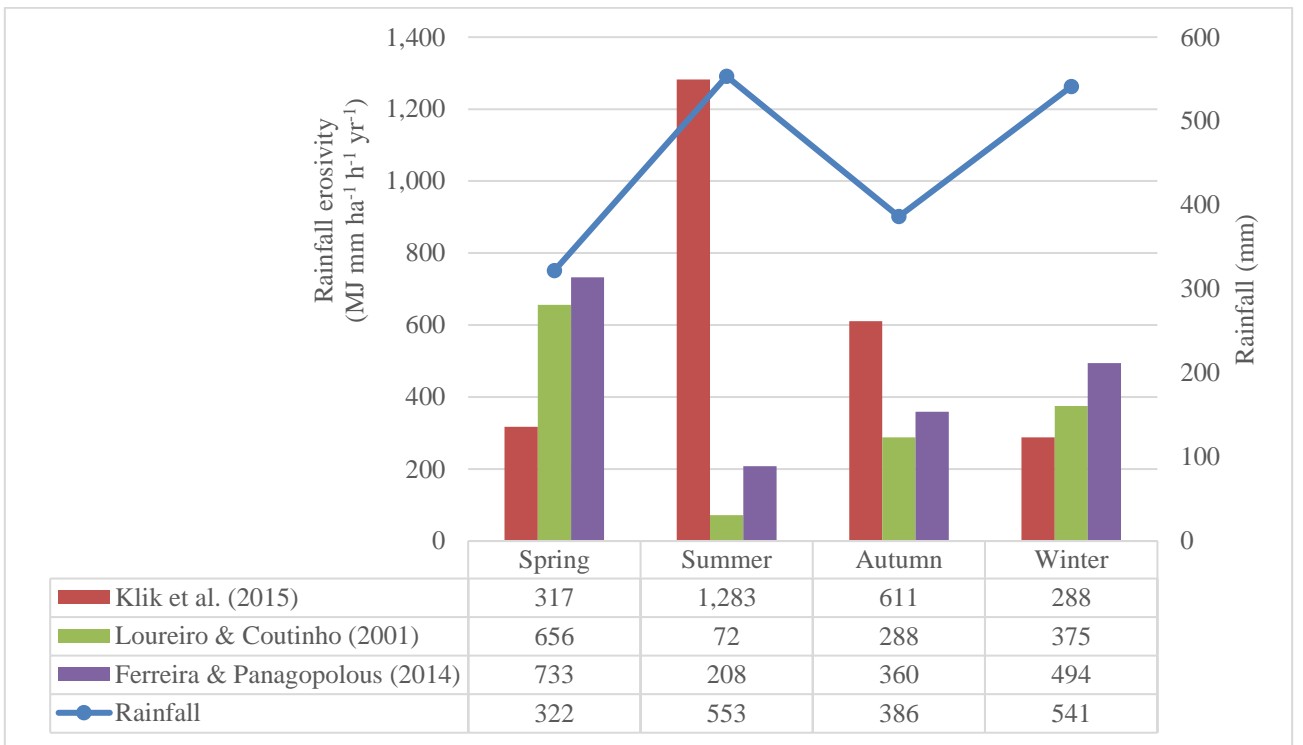

| | Spring | Summer | Autumn | Winter |
|---|---|---|---|---|
| Klik et al. (2015) | 317 | 1,283 | 611 | 288 |
| Loureiro & Coutinho (2001) | 656 | 72 | 288 | 375 |
| Ferreira & Panagopolous (2014) | 733 | 208 | 360 | 494 |
| Rainfall | 322 | 553 | 386 | 541 |

**Figure 1: Graph of seasonal rainfall and estimates of erosivity in the Mangatarere.**

**Table 1: Annual estimates of erosivity in the Mangatarere (MJ mm ha$^{-1}$ h$^{-1}$ yr$^{-1}$).**

| Equation Source | Klik et al. (2015) | Loureiro & Coutinho (2001) | Ferreira & Panagopolous (2014) |
|---|---|---|---|
| **Annual erosivity** | 2607 | 1391 | 1715 |

**Table 2: Summary of different studies that developed rainfall erosivity equations, original locations, and other studies that used their equations.**

| # | Author | Original Location | Resolution | Equation and requirements | Other studies |
|---|---|---|---|---|---|
| 1 | Wischmeier and Smith (1978) and Renard et al. (1997) | United States of America | Sub-daily | $$R = \frac{\sum_{i=1}^{j}(EI_{30})_i}{N}$$ $$EI_{30} = E \times I_{30}$$ $$E = 916 + 331 \times log_{10} I$$ <br><br> I = intensity (in/hr) <br> $EI_{30i}$ = $EI_{30}$ for storm i <br> j = number of storms in an N-year period <br><br> Units <br> Imperial: <br> Hundreds of foot • tonf • inch • acre$^{-1}$ • hour$^{-1}$ • year-1 <br><br> Metric (multiply by 17.02): <br> Megajoule •millimetre • hectare$^{-1}$ • hour$^{-1}$ • year$^{-1}$ | Applied around USA |
| 2 | Mihara (1951) and Hudson (1971) as cited in David (1988) | USA | Daily | $$R = A \times \sum_{1}^{n} P_i^m$$ A = 0.002 <br> M = 2 <br> $P_i$ = Precipitation total for day $i$ when $P$ exceeds 25mm <br><br> Units: Not specified, likely to be original USLE imperial units | Watersheds around the Philippines (David, 1988) |
| 3 | Arnoldus (1980) as cited in Renard and Freimund (1994) | Morocco and other locations in West Africa | Monthly and annual | West Africa <br> $$R = 4.79 MFI - 142$$ $$R = 5.44 MFI - 416$$ Eastern USA <br> $$R = 6.86 MFI - 420$$ Western USA <br> $$R = 4.79 MFI - 143$$ Northwest USA <br> $$R = 0.66 MFI - 3$$ <br><br> $$MFI = \sum_{i=1}^{12} \frac{P_i^2}{P}$$ | Morocco Turkey (Demirci & Karaburun, 2012); Morocco (Raissouni et al., 2016) |

| | | | | | |
|---|---|---|---|---|---|
| | | | MFI = Modified Fournier's Index<br>$P_i$ = monthly precipitation<br>$P$ = annual precipitation | | |
| 4 | Renard and Freimund (1994) | West coast of USA | Monthly and annual | Units:<br>Ton-metre • centimetre • hectare$^{-1}$ • hour$^{-1}$ • year$^{-1}$ (Renard and Freimund, 1994)<br>$$R = 0.0483 \times P^{1.610}$$<br>$$R = 587.8 - 1.219P + 0.004105P^2$$<br><br>Using MFI (Arnoldus, 1980):<br>$$R = 0.07397 \times MFI^{1.847}$$<br>$$R = 95.77 - 6.081MFI + 0.4770MFI^2$$<br>$P_i$ = monthly precipitation<br>$P$ = annual precipitation | Central America (Kim et al., 2005) Iran (Zakerinejad & Maerker, 2015) Ecuador (Ochoa-Cueva et al., 2015) |
| 5 | Zhou et al. (1995) as cited in Li et al. (2014) | Southern China | Monthly | Units: Megajoule •millimetre • hectare$^{-1}$ • hour$^{-1}$ • year$^{-1}$<br>$$R = \sum_{i=1}^{12} -1.15527 + 1.792P_i$$<br>$P_i$ = monthly precipitation | China (Li et al., 2014) |
| 6 | Roose (1975) and Morgan (1974) as cited in Morgan (2005) | Peninsular Malaysia and Africa | Annual | Units: Megajoule •millimetre • hectare$^{-1}$ • hour$^{-1}$ • year$^{-1}$<br>Africa (Roose, 1975):<br>$$R = 0.5 \times P \times 17.3$$<br>Peninsular Malaysia:<br>$$R = (9.28 \times P - 8838)\left(\frac{75}{1000}\right)$$<br>$P$ = mean annual precipitation (mm) | Malaysia (Roslee et al., 2017); Vanuatu (Dumas & Fossey, 2009); Iran (Zakerinejad & Maerker, 2015) |
| 7 | El-Swaify et al. (1987) as cited in Merritt et al. (2004) | Possibly Thailand | Annual | Units: Megajoule •millimetre • hectare$^{-1}$ • hour$^{-1}$ • year$^{-1}$<br>$$R = 38.5 + 0.35P$$<br>$P$ = mean annual precipitation<br><br>Units: Tons • hectare$^{-1}$ • year$^{-1}$ (All the other factors must have been developed to have no units so that the final soil loss is in tons/ha/year) | Thailand (Eiumnoh, 2000; Merritt et al., 2004); Philippines (Adornado & Yoshida, 2010; Adornado et al., 2009; Hernandez et al., 2012); Sri Lanka (Jayasinghe et al., 2010) |
| 8 | Land Development Department (2000), as cited in Nontananandh | Thailand | Annual | Units: Megajoule •millimetre • hectare$^{-1}$ • hour$^{-1}$ • year$^{-1}$<br>$$R = 0.04669P - 12.1415$$<br>$P$ = mean annual rainfall<br><br>Units: Megajoule •millimetre • hectare$^{-1}$ • hour$^{-1}$ • year$^{-1}$ | Thailand (Nontananandh & Changnoi, 2012) |

| # | Reference | Country | Time scale | Equation | Applied in |
|---|-----------|---------|-----------|----------|-----------|
| | and Changnoi (2012) | | | | |
| 9 | Loureiro and Coutinho (2001) | Portugal | Daily | $$R = \frac{1}{N}\sum_{i=1}^{N}\sum_{m=1}^{12} EI_{30(monthly)}$$ $$EI_{30\,(monthly)} = 7.05 rain_{10} - 88.92 days_{10}$$ $Rain_{10}$ = monthly rainfall for days with $\geq$ 10.0mm of rain<br>$Days_{10}$ = monthly number of days with rainfall $\geq$ 10.0mm of rain<br>N = number of years<br><br>Units: Megajoule •millimetre • hectare$^{-1}$ • hour$^{-1}$ • year$^{-1}$ | Spain (López-Vicente, Navas, & Machín, 2008) |
| 10 | Fernandez et al. (2003), originally developed by the USDA-ARS (2002) | USA | Annual | $$R = -823.8 + 5.213P$$ P = annual precipitation<br><br>Units: Megajoule •millimetre • hectare$^{-1}$ • hour$^{-1}$ • year$^{-1}$ | USA (Fernandez et al., 2003); Greece (Jahun et al., 2015) |
| 11 | Ram et al. (2004), as cited in Jain and Das (2010) | India | Annual | $$R = 81.5 + 0.38P$$ P = annual precipitation for areas where annual precipitation ranges between 340mm to 3500mm<br><br>Units: Megajoule •millimetre • hectare$^{-1}$ • hour$^{-1}$ • year$^{-1}$ | India (Jain & Das, 2010) |
| 12 | Shamshad et al. (2008) | Malaysia | Monthly and annual | Based on Loureiro and Coutinho (2001) but for Malaysia:<br>$$R = \sum_{i=1}^{12} 6.97 rain_{10} - 11.23 days_{10}$$ $$R = \sum_{i=1}^{12} 0.266 \times rain_{10}^{2.071} \times days_{10}^{-1.367}$$ $$R = \sum_{i=1}^{12} 227 \times \left(\frac{P_i^2}{P}\right)^{0.548}$$ $Rain_{10}$ = monthly rainfall for days with $\geq$ 10.0mm of rain<br>$Days_{10}$ = monthly number of days with rainfall $\geq$ 10.0mm of rain<br>$P_i$ = monthly precipitation<br>P = annual precipitation<br><br>Units: Megajoule •millimetre • hectare$^{-1}$ • hour$^{-1}$ • year$^{-1}$ | Philippines (Delgado & Canters, 2012) |
| 13 | Irvem et al. (2007) | Turkey | Monthly and annual | $$R = 0.1215 \times MFI^{2.2421}$$ | Turkey (Ozsoy et al., 2012) |

$$MFI = \sum_{i=1}^{12} \frac{P_i^2}{P}$$

$P_i$ = monthly precipitation
$P$ = annual precipitation

Units: Megajoule •millimetre • hectare$^{-1}$ • hour$^{-1}$ • year$^{-1}$

| 14 | Ferreira and Panagopolous (2014), similar to Loureiro and Coutinho (2001) | Portugal | Daily | $$R = \sum_{i=1}^{12} 6.56 rain_{10} - 75.09 days_{10}$$ Rain$_{10}$ = monthly rainfall for days with $\geq$ 10.0mm of rain <br> Days$_{10}$ = monthly number of days with rainfall $\geq$ 10.0mm of rain <br><br> Units: Megajoule •millimetre • hectare$^{-1}$ • hour$^{-1}$ • year$^{-1}$ | Portugal (Ferreira & Panagopoulos, 2014) |
|---|---|---|---|---|---|
| 15 | Nakil (2014) as cited in Nakil and Khire (2016) | India | Annual | $$R = 839.15 \times e^{0.0008P}$$ $P$ = annual precipitation <br><br> Units: Megajoule •millimetre • hectare$^{-1}$ • hour$^{-1}$ • year$^{-1}$ | India (Nakil & Khire, 2016) |
| 18 | Naipal et al. (2015) | Global application, but original data from USA and Europe | Annual | Various equations depending on Köppen climate classification, including alternate equations if SDII is not available <br><br> $P$ = annual precipitation (mm) <br> $Z$ = mean elevation (m) <br> SDII = simple precipitation intensity index (mm day$^{-1}$) | |
| 19 | Klik et al. (2015) | New Zealand | Annual or seasonal | Annual or seasonal: $$R = aP^b$$ $$R = aP + b$$ $P$ = annual precipitation (mm) or seasonal precipitation (mm) <br> a & b = constants depending on region of New Zealand <br><br> The equation used will depend on the region of New Zealand, and the season. <br><br> Units: Megajoule •millimetre • hectare$^{-1}$ • hour$^{-1}$ | |
| 20 | Sholagberu et al. (2016) | Malaysia | Annual | $$R = 0.0003P^{1.771}$$ $P$ = annual precipitation <br><br> Units: Megajoule •millimetre • hectare$^{-1}$ • hour$^{-1}$ • year$^{-1}$ | |

**Table 3: Summary of different studies with soil erodibility equations, original locations, and other studies that used their equations. All of the equations in Table 2 use imperial units of soil erodibility: ton • acre • hour • hundreds of acre$^{-1}$ • foot$^{-1}$ • tonf$^{-1}$ • inch$^{-1}$. Multiply by 0.1317 to give in SI units of metric ton • hectare • hour • hectare$^{-1}$ • megajoule$^{-1}$ • millimetre$^{-1}$.**

| # | Author | Original Location | Data requirements | Equation | Other studies |
|---|--------|-------------------|-------------------|----------|---------------|
| 1 | Wischmeier and Smith (1978) and Renard et al. (1997) | USA | Very fine sand (%), clay (%), silt (%), organic matter (%), soil structure, profile-permeability | $$M = Silt \times (100 - Clay)$$ $$K = \{[2.1 \times M^{1.14} \times (10^{-4}) \times (12 - a)]$$ $$+ [3.25 \times (b - 2)]$$ $$+ [2.5 \times (c - 3)]\} \div 100$$ <br><br> M = Particle-size parameter <br> Silt = Silt (%) but also includes the percentage of very fine said (0.1 to 0.05mm) <br> Clay = Clay (%) <br> a = Organic matter (%) <br> b = Soil-structure code used in soil classification: <br> • 1: Very fine granular <br> • 2: Fine granular <br> • 3: Medium or coarse granular <br> • 4: Blocky, platy, or massive <br> c = Profile-permeability class <br> • 1: Rapid <br> • 2: Moderate to rapid <br> • 3: Moderate <br> • 4: Slow to moderate <br> • 5: Slow <br> • 6: Very slow | Thailand (Eiumnoh, 2000); Vanuatu (Dumas & Fossey, 2009); Philippines (Schmitt, 2009); India (Jain & Das, 2010); Turkey (Ozsoy et al., 2012); Iran (Bagherzadeh, 2014); Portugal (Ferreira & Panagopoulos, 2014); China (Li et al., 2014); European Union (Panagos et al., 2014) |
| 2 | Williams and Renard (1983) as cited in Chen et al. (2011) | USA | Sand (%), silt (%), clay (%), organic carbon (%) | $$K = 0.2 + 0.3 \exp\left(0.0256 \times Sa \times \left(1 - \frac{Si}{100}\right)\right)$$ $$\times \left(\frac{Si}{Cl + Si}\right)^{0.3}$$ $$\times \left(1.0 - \frac{0.25 \times C}{C + \exp(3.72 - 2.95C)}\right)$$ $$\times \left(1.0 - \frac{0.7 \times SN}{SN + \exp(-5.51 + 22.9SN)}\right)$$ <br> Sa = Sand % <br> Si = Silt % <br> Cl = Clay % <br> SN = 1-(Sa/100) <br> C = Organic Carbon | China (Chen et al., 2011) |
| 3 | David (1988), a simplified | USA | Sand (%), clay (%), silt (%), organic | $$K = [(0.043 \times pH) + (0.62 \div OM) + (0.0082 \times S)$$ $$- (0.0062 \times C)] \times Si$$ | Philippines (David, 1988; |

| | | | version of Wischmeier and Mannering (1969) | matter (%), pH | pH = pH of the soil OM = Organic matter in percent S = Sand content in percent C = Clay ratio = % clay / (% sand + % silt) Si = Silt content = % silt / 100 | Hernandez et al., 2012) |

| # | Author | Original Location | Data | Equation | |
|---|--------|-------------------|------|----------|---|

version of Wischmeier and Mannering (1969) — matter (%), pH

pH = pH of the soil
OM = Organic matter in percent
S = Sand content in percent
C = Clay ratio = % clay / (% sand + % silt)
Si = Silt content = % silt / 100

Hernandez et al., 2012)

4 El-Swaify & Dangler (1976) as cited in Renard et al. (1997) — Hawaii, USA — Textural information, base saturation

$$K = -0.03970 + 0.00311x_1 + 0.00043x_2 + 0.00185x_3 + 0.00258x_4 - 0.00823x_5$$

$x_1$ = unstable aggregate size fraction (<0.250mm) (%)
$x_2$ = modified silt (0.002 - 0.1mm) (%) * modified sand (0.1 - 2mm) (%)
$x_3$ = % base saturation
$x_4$ = silt fraction (0.002 - 0.050mm) (%)
$x_5$ = modified sand fraction (0.1 - 2mm) (%)

**Table 4: K-factor values from Dymond (2010) for soil textures in New Zealand.**

| Soil Texture | K-factor (Dymond, 2010) |
|--------------|------------------------:|
| Clay | 0.20 |
| Loam | 0.25 |
| Sand | 0.05 |
| Silt | 0.35 |

**Table 5: K-factor values from David (1988) for soil textures in the Philippines.**

| Soil Texture | K-factor (David, 1988) |
|--------------|-----------------------:|
| Loamy fine sand | 0.07 |
| Clay | 0.13–0.26 |
| Clay loam | 0.22–0.30 |
| Loam | 0.19–0.63 |
| Sandy clay | 0.09–0.20 |
| Sandy loam | 0.23–0.30 |
| Silt loam | 0.30–0.60 |
| Silty clay | 0.19–0.27 |
| Silty clay loam | 0.28–0.35 |

**Table 6: Summary of methods of calculating LS-factor, original locations, and other studies that used these methods.**

| # | Author | Original Location | Data requirements | Equation | Other studies that utilised similar methods |
|---|--------|-------------------|-------------------|----------|---------------------------------------------|
| 1 | Wischmeier and Smith (1978) | USA | Slope length and angle | $LS = (\frac{\lambda}{72.6})^m \times [(65.41 \times \sin^2\theta) + (4.56 \times \sin\theta) + 0.065]$ | Thailand (Eiumnoh, 2000; Merritt et al., 2004); Vanuatu (Dumas |

| | | | | | & Fossey, 2009); Iran (Bagherzadeh, 2014) |
|---|---|---|---|---|---|
| | | | λ = Slope length in feet<br>Θ = Angle of slope<br>m = Dependent on the slope<br>  • 0.5 if slope > 5%<br>  • 0.4 if slope is between 3.5% and 4.5%<br>  • 0.3 if slope is between 1% and 3%<br>  • 0.2 if slope is less than 1% | | |
| 2 | Renard et al. (1997) | USA | Slope length and angle | $$L = \left(\frac{\lambda}{72.6}\right)^m$$ $$m = \frac{\beta}{1+\beta}$$ $$\beta = \frac{(\frac{\sin\theta}{0.0896})}{[3.0 \times (\sin\theta)^{0.8} + 0.56]}$$<br><br>If slope is less than 9%:<br>$$S = 10.8 \times \sin\theta + 0.03$$<br><br>If slope is greater or equal to 9%:<br>$$S = 16.8 \times \sin\theta - 0.50$$<br><br>But if the slope is shorter than 15 feet:<br>$$S = 3.0 \times (\sin\theta)^{0.8} + 0.56$$<br><br>λ = Slope length in feet<br>Θ = Angle of slope<br>m = Dependent on the slope<br>  • 0.5 if slope > 5%<br>  • 0.4 if slope is between 3.5% and 4.5%<br>  • 0.3 if slope is between 1% and 3%<br>  • 0.2 if slope is less than 1% | Philippines (Schmitt, 2009); China (Li et al., 2014); Thailand (Nontananandh & Changnoi, 2012); Turkey (Ozsoy et al., 2012) |
| 3 | David (1988), based on work by Madarcos (1985) and Smith & Whitt (1947) | Philippines, but based on work from the USA | Slope rise in percent | $$LS = a + b \times S_L^{4/3}$$<br><br>a = 0.1<br>b = 0.21<br>$S_L$ = Slope in percent | Philippines (David, 1988) |
| 4 | Morgan (2005) but previously published in earlier editions | Britain | Slope length and gradient in percent | $$LS = \left(\frac{l}{22}\right)^{0.5}(0.065 + 0.045s + 0.0065s^2)$$<br><br>l = slope length (m) | India (Nakil & Khire, 2016; Sinha & Joshi, 2012); Greece (Rozos et al., 2013) |

| 5 | Moore & Burch (1986) as cited in Mitasova et al. (1996); Desmet & Govers (1996); Mitasova et al. (2013); | USA | Upslope contributing area per unit width, which can be approximated through flow accumulation, cell size, slope | $s$ = slope steepness (%) $$LS = (m + 1)\left(\frac{U}{L_0}\right)^m \left(\frac{\sin \beta}{S_0}\right)^n$$ U (m²m⁻¹) = upslope contributing area per unit width as a proxy for discharge $$U = Flow\ Accumulation\ \times Cell\ Size$$ $L_0$ = length of the unit plot (22.1) $S_0$ = slope of unit plot (0.09) $\beta$ = slope m (sheet) and n (rill) depend on the prevailing type of erosion (m= 0.4 to 0.6) and n (1.0 to 1.3) | Philippines (Adornado & Yoshida, 2010; Adornado et al., 2009); Sri Lanka (Jayasinghe et al., 2010); China (Chen et al., 2011); Iran (Zakerinejad & Maerker, 2015); Jordan (Farhan & Nawaiseh, 2015); Morocco (Raissouni et al., 2016); New Zealand (Fernandez & Daigneault, 2016) Similar methods from Moore & Burch (1986): India (Jain & Das, 2010); Portugal (Ferreira & Panagopoulos, 2014); Greece (Jahun et al., 2015); India (Nakil & Khire, 2016) Similar methods from Desmet & Govers (1996): USA (Boyle et al., 2011); Turkey (Demirci & Karaburun, 2012); Philippines (Delgado & Canters, 2012) |
|---|---|---|---|---|---|

**Table 7: C-factor equations that use NDVI.**

| # | Author | Original Location | Equation |
|---|--------|-------------------|----------|
| 1 | Van der Knijff et al. (2000) | Europe | $$C = \exp\left[-\propto \left(\frac{NDVI}{\beta - NDVI}\right)\right]$$ $\alpha = 2$ $\beta = 1$ |
| 2 | Ma et al. (2001) as cited in Li et al. (2014) | China | $$f_g = \frac{NDVI - NDVI_{min}}{NDVI_{max} - NDVI_{min}}$$ $$C = \begin{cases} 1 & f_g = 0 \\ 0.6508 - 0.343 \times \log(f_g) & 0 < f_g < 78.3\% \\ 0 & f_g \geq 78.3\% \end{cases}$$ |

**Table 8: C-factors for general types of land cover compiled from various sources.**

| Cover | Dymond (2010) (New Zealand) | David (1988) (Philippines) | Morgan (2005) (Various) | Fernandez et al. (2003) (USA) | Dumas & Fossey (2009) (Vanuatu) | Land Development Department (2002) as cited in Nontananandh & Changnoi (2012) |
|---|---|---|---|---|---|---|
| Bare ground | 1 | 1 | 1 | | | |
| Urban | | 0.2 | | 0.03 | 0 | 0 |
| Crop | | | | 0.128 | 0.01 | 0.255–0.525 |
| Forest | 0.005 | 0.001–0.006 | 0.001 | 0.001 | 0.001 | 0.003–0.048 |
| Pasture | 0.01 | | 0.1 | | | |
| Scrub | 0.005 | 0.007–0.9 | 0.01 | 0.003 | 0.16 | 0.01–0.1 |

**Table 9: C-factors for specific types of land cover compiled from various sources.**

| Cover | Panagos et al. (2015b) (Europe) | David (1988) (Philippines) | Morgan (2005) (Various) |
|---|---|---|---|
| Bananas | | 0.1–0.3 | |
| Barley | 0.21 | | |
| Chili | | | 0.33 |
| Cocoa | | | 0.1–0.3 |
| Coffee | | | 0.1–0.3 |
| Common wheat and spelt | 0.2 | | 0.1–0.4 |
| Cotton seed | 0.5 | 0.4–0.6 | 0.4–0.7 |
| Dried pulses (legumes) and protein crop | 0.32 | 0.3–0.5 | 0.04–0.7 |
| Durum wheat | 0.2 | | |
| Fallow land | 0.5 | | |
| Grain maize-corn | 0.38 | 0.3–0.6 | 0.02–0.9 |
| Groundnuts | | | 0.3–0.8 |
| Linseed | 0.25 | | 0.1–0.2 |
| Oilseeds | 0.28 | | |
| Palm with cover crops | | 0.05–0.3 | 0.1–0.3 |
| Pineapple | | 0.2–0.5 | 0.01–0.4 |
| Potatoes | 0.34 | | 0.1–0.4 |
| Rape and turnip rape | 0.3 | | |
| Rice | 0.15 | 0.1–0.2 | 0.1–0.2 |
| Rye | 0.2 | | |
| Soya | 0.28 | | 0.2–0.5 |
| Sugar beet | 0.34 | | |
| Sugarcane | | | 0.13–0.4 |
| Sunflower seed | 0.32 | | |
| Tobacco | 0.49 | 0.4–0.6 | |

| Yams | | 0.4–0.5 |
|---|---|---|

**Table 10: Examples of where C-factor accounts for crop management from Morgan (2005) and David (1988).**

| Crop | Management | C-factor |
|---|---|---|
| Maize, sorghum or millet | High productivity; conventional tillage | 0.20–0.55 |
| | Low productivity; conventional tillage | 0.50–0.90 |
| | High productivity; chisel ploughing into residue | 0.12–0.20 |
| | Low productivity; chisel ploughing into residue | 0.30–0.45 |
| | High productivity; no or minimum tillage | 0.02–0.10 |
| Coconuts | Tree intercrops | 0.05–0.1 |
| | Annual crops as intercrop | 0.1–0.30 |

**Table 11: P-factors for different types of agricultural management practices.**

| | David (1988) | | | | |
|---|---|---|---|---|---|
| **Tillage and Residue Management** | **P-factor** | | | | |
| Conventional tillage | 1.00 | | | | |
| Zoned tillage | 0.25 | | | | |
| Mulch tillage | 0.26 | | | | |
| Minimum tillage | 0.52 | | | | |
| **Slope (%)** | **Terracing** | | **Contouring** | **Contour Strip Cropping** | |
| | **Bench** | **Broad-based** | | | |
| 1 – 2 | 0.10 | 0.12 | 0.60 | 0.30 | |
| 3 – 8 | 0.10 | 0.10 | 0.50 | 0.15 | |
| 9 – 12 | 0.10 | 0.12 | 0.60 | 0.30 | |
| 13 – 16 | 0.10 | 0.14 | 0.70 | 0.35 | |
| 17 – 20 | 0.12 | 0.16 | 0.80 | 0.40 | |
| 21 – 25 | 0.12 | 0.18 | 0.90 | 0.45 | |
| > 25 | 0.14 | 0.20 | 0.95 | 0.50 | |
| **Panagos et al. (2015c)** | | | | | |
| **Slope (%)** | **Contouring P-factor** | | | | |
| 9 – 12 | 0.6 | | | | |
| 13 – 16 | 0.7 | | | | |
| 17 – 20 | 0.8 | | | | |
| 21 – 25 | 0.9 | | | | |
| > 25 | 0.95 | | | | |

**Table A1: Summary of previous studies that have applied the USLE and RUSLE**

| Author | Location | R-factor | K-factor | LS-factor | C-factor | P-factor |
|---|---|---|---|---|---|---|
| David (1988) | Various watersheds in the Philippines | Mihara (1951) and Hudson (1971) as cited in David (1988) | Wischmeier and Mannering (1969) | Madarcos (1985) and Smith & Whitt (1947) | Literature | Literature |

| | | | | | | |
|---|---|---|---|---|---|---|
| Eiumnoh (2000) | Sakae Krang watershed (Thailand) | El-Swaify et al. (1987) as cited in Merritt et al. (2004) | USLE method | USLE method | Literature | None observed (P=1) |
| Fernandez et al. (2003) | Lawyers Creek Watershed (USA) | USDA-ARS (2002) | From the SSURGO database (USDA) | Upslope contributing area method | Database from RUSLE software | Database from RUSLE software |
| Merritt et al. (2004) | Mae Chem watershed (Thailand) | El-Swaify et al. (1987) as cited in Merritt et al. (2004) | Previous studies in area | USLE method | Previous studies in area | Previous studies in area |
| Post and Hartcher (2005) | Mae Chem watershed (Thailand) | El-Swaify et al. (1987) as cited in Merritt et al. (2004) | Previous studies in area | L = 1 S = derived from DEM | Previous studies in area | None observed (P=1) |
| Dumas and Fossey (2009) | Efate Island (Vanuatu) | Roose (1975) and Morgan (1994) as cited in Morgan (2005) | USLE method | RUSLE method at pixel level | Literature | None observed (P=1) |
| Adornado et al. (2009) | REINA (Philippines) | El-Swaify et al. (1987) as cited in Merritt et al. (2004) | Table by Stewart et al. (1975) | Upslope contributing area method | Literature | None observed (P=1) |
| Schmitt (2009) | Negros Island (Philippines) | RUSLE method | USLE method | RUSLE method at pixel level | Literature | Previous studies |
| Jayasinghe et al. (2010) | Nuwaraeliya (Sri Lanka) | El-Swaify et al. (1987) as cited in Merritt et al. (2004) | Table by Stewart et al. (1975) | Upslope contributing area method | Literature | None observed (P=1) |
| Jain and Das (2010) | Jharkhand (India) | Ram et al. (2004), as cited in Jain and Das (2010) | USLE method and previous studies | Upslope contributing area method | Literature | None observed (P=1) |
| Adornado and Yoshida (2010) | Bukidnon (Philippines) and also REINA (Philippines) | El-Swaify et al. (1987) as cited in Merritt et al. (2004) | Table by Stewart et al. (1975) | Upslope contributing area method | Literature | None observed (P=1) |
| Boyle et al. (2011) | California (USA) | From previous studies | From previous studies | Upslope contributing area method | Literature | N/A |
| Chen et al. (2011) | Xiangxi watershed (China) | Wischmeier and Smith (1978) | Williams and Renard (1983) nomograph | Upslope contributing area method | Using NDVI | N/A |
| Demirci & Karaburun (2012) | Buyukcekmece Lake watershed (Turkey) | Arnoldus (1980) | Torri et al. (1997) equation | Upslope contributing area method | Using NDVI | None observed (P=1) |
| Nontananandh and Changnoi (2012) | Songkhran watershed (Thailand) | Land Development Department (2000) | Values from Land Development Department (2000) | Modified RUSLE method | Literature | None observed (P=1) |

| | | | | | | |
|---|---|---|---|---|---|---|
| Ozsoy et al. (2012) | Mustafakemalpasa River Basin (Turkey) | From previous studies | USLE method | RUSLE method, using a 3rd party programme | Literature | None observed (P=1) |
| Delgado & Canters (2012) | Claveria (Philippines) | Shamshad et al. (2008) | USLE method | RUSLE2 programme, using the upslope contributing area method | Literature | David (1988) |
| Hernandez et al. (2012) (used SedNet, which has an USLE component) | Pagsanjan (Philippines) | El-Swaify et al. (1987) as cited in Merritt et al. (2004) | Wischmeier and Mannering (1969) | Algorithm within SedNet | Literature | N/A |
| Sinha & Joshi (2012) | Maharashtra (India) | Roose (1975) | USLE method | Morgan (1986) | Literature | Literature |
| Nigel & Rughooputh (2012) | Mauritius | Arnoldus (1980), as cited in Le Roux et al. (2005) | From previous studies | Upslope contributing area method | Literature | Literature |
| Životić et al. (2012) | Nisava river basin (Serbia) | Wischmeier and Smith (1978) | USLE method | RUSLE method | Using NDVI | None observed (P=1) |
| Rozos et al. (2013) | Euboea Island (Greece) | Flabouris (2008) | Based on geological characteristics | Morgan (1986) | Literature | None observed (P=1) |
| Bagherzadeh (2014) | Masshad plain (Iran) | Wischmeier and Smith (1978) | USLE method | USLE method | | None observed (P=1) |
| Ferreira and Panagopoulos (2014) | Alqueva (Portugal) | Similar to Loureiro and Coutinho (2001) | USLE method | Upslope contributing area method | Using NDVI | None observed (P=1) |
| Li et al. (2014) | Guangdong (China) | Zhou et al. (1995) | USLE method | Similar to RUSLE method | Using NDVI | 1 for wasteland and built-up 0.5 for forested 0.2 for orchard land 0.35 for cropland |
| Zakerinejad and Maerker (2015) (used USPED, which has USLE components) | Mazayjan (Iran) | Ferro et al. (1991); Renard & Freimund (1994); Sadeghifard et al. (2004) | RUSLE method | Algorithm within USPED | Literature | None observed (P=1) |
| Jahun et al. (2015) | Crete (Greece) | Fu et al. (2006) | RUSLE method | Upslope contributing area method | Using NDVI | Previous studies |
| Farhan and Nawaiseh (2015) | Wadi Kerak catchment (Jordan) | Eltaif et al. (2010) | Similar to USLE nomograph | Upslope contributing area method | Literature | Literature |
| Panagos et al. (2015e) and related papers | Europe | Rainfall Intensity Summarisation Tool (RIST) | USLE method | 3rd party programme | Literature | Literature |

| | | | | | | |
|---|---|---|---|---|---|---|
| Russo (2015) | Brunei Darussalam | Rosewell & Turner (1992) | Rosewell (1997) | RUSLE method | Based on ground covered | None observed (P=1) |
| Nakil and Khire (2016) | Gangapur (India) | Nakil (2014) | USLE method | RUSLE method | Literature | Literature |
| Raissouni et al. (2016) | Smir Dam (Morocco) | Similar to Arnoldus (1980) methods | Merzouk (1985) | Upslope contributing area method | Literature | None observed (P=1) |
| Fernandez and Daigneault (2016) | Waikato (New Zealand) | Institute of Water Research (2015) | Dymond et al. (2010) | Upslope contributing area method | Range between 1 (wood vegetation) and 10 (herbaceous vegetation or bare ground) | |
| Duarte et al. (2016) | Montalegre (Portugal) | Loureiro and Coutinho (2001) | USLE method | USLE method | Literature | Literature |
| Gaubi et al. (2017) | Lebna watershed (Tunisia) | Rango and Arnoldus (1987) | USLE method | Upslope contributing area method | Literature | None observed (P=1) |