# Peer review of "A review of the (Revised) Universal Soil Loss Equation (R/USLE): with a view to increasing its global applicability and improving soil loss estimates"

_Hydrology and Earth System Sciences, 2018_

## Referee Comment (RC1) · Anonymous Referee #1 · 15 May 2018

**Referee comment to:**

Journal: HESS

A review of the (Revised) Universal Soil Loss Equation (R/USLE): with a view to increasing its global applicability and improving soil loss estimates

Author(s): R. Benavidez et al.

Hydrol. Earth Syst. Sci. Discuss., hess-2018-68

**General comments:**

This review paper presents a comprehensive overview of studies applying the R/ULSE all over the world and provides information on how different studies have adapted the equations to calculate the factors of the USLE to local conditions. In addition, studies dealing with limitations of the USLE and future developments of the approach are mentioned. The authors explain that they provided this review to serve as a reference for other researchers working with the USLE.

In general, a review of the USLE is well placed in HESS. The authors have done a very diligent work by summarizing many publications applying the USLE. In addition the manuscript provides some helpful hints, as for example the advice to be careful with the units of the USLE-factors used in different studies (i.e. for the K-factor in Chapter 2.2).

However, my major objection is, that the manuscript provides only an overview of existing studies and that a critical examination of the approaches presented in the manuscript is missing. Thus, I cannot see a significant own contribution of the authors besides the summary of existing studies on the application of the USLE. The manuscript should thus be thoroughly revised and provide a critical analysis of the approaches presented to gain new insight in the topic. Further comments for revision of the manuscript are given in the following:

- The introduction is very general. It should be worked out, why this review of the USLE is necessary and what is its benefit in relation to other reviews. In addition, the objectives are not clear and included at various locations in the introduction.
  Thus, the introduction should clearly motivate this review, leading to focused objectives at the end of the introduction (see also specific comments).

- The authors promise, that they will provide guidance which equation is most appropriate for a range of different geoclimatic regions (Page 2, line 17 – 18). However, the advices are very general and the studies presented in Chapter 2 seem to be randomly picked. For example, Chapter 2.1 provides a comprehensive overview of 19 studies that have derived approaches to estimate R-factors for different regions or have applied these approaches (Table 3). Furthermore, the authors summarize various studies using approaches to calculate R-factors in regions other than those for which they were developed. Following this, a simple calculation example is provided (Page 6, line 8 – 16 and Figure 1): In this example, 2 equations developed for Portugal and 1 equation developed for New Zealand are applied to a watershed in New Zealand (Figure 1). As expected, the equations developed for Portugal do not match the seasonal variation in New Zealand. The authors conclude that it is important to understand the regional

applicability of rainfall erosivity equations (Page 6, line 17-18). Although many studies were reviewed, the main result of Chapter 2.1 is a very general statement drawn on the basis of a simple example.

If such examples are provided, they should cover a much larger number of approaches and data of different regions to derive useful conclusions to guide other users of the USLE. It would be much more important to analyze, if approaches for R-factors could be transferred to regions with similar climate characteristics for which no detailed data is available and what criteria should be applied to do this.

- In Chapter 2.2 only studies for the US are presented. It would be interesting, how studies in other regions deal with K-factors?

- Chapter 3 is about limitations of the R/USLE. As before, only existing studies dealing with the limitations of the USLE are summarized and a critical analysis of the limitations is missing (see also specific comments).
  The topic of validation of estimated soil loss rates by using the USLE is mentioned only briefly. In my opinion, it is one of the major limitations of the USLE that it is so difficult to validate the estimated soil loss rates. This topic should be discussed in more detail.

- In Chapter 4 again only studies are summarized which are dealing with further developments of the USLE, but again, a critical analysis is missing.

- The conclusions are very general.

- The abstract is very brief. It should be thoroughly revised according to the revision of the manuscript.

**Specific comments:**

- Page 1, line 22 – Page 2, line 5: The introductory part on soil erosion is very long and not specific for the USLE. It should be shortened and focused.

- Page 2, line 6 – 13: In this section, a few review papers on erosion models are presented. It is not clear, why these reviews have been selected. I suggest to focus on previous reviews of the USLE and to work out, why the additional review presented in this paper is necessary and what will be the benefit of it.

- Page 2, line 15 – 19: I suggest to move this section to the objectives at the end of the introduction.

- Page 2, line 28 – 29: move to objectives at the end of the introduction. In addition, it should be made clear, which limitations of the USLE are analyzed.

- Page 2, line 30 – 34: redundant to the section above. Include the information not yet provided in line 19 – 27 into this section.

- Page 3, line 10 – 13: The objectives of the study mentioned at various locations in the introduction should be summarized at the end of the introduction (see comments above).

- Page 3, line 19 – 26: In my opinion, this information fits better in the introduction.

- Page 4, line 1 – 5: some additional objectives are mentioned in this section → should be moved to a focused section presenting the objectives at the end of the introduction.

- Page 3, Chapter 2: Some general information on the USLE should be provided, i.e. that it was developed from soil loss rates on plot experiments.

- Page 11, line 6: the information on the R/ULSE unit plot is also essential for the other factors. It should be mentioned in the preface of Chapter 2, i.e. page 3, line 19 - 26.

- Page 20, line 2 – 10: in this paragraph it is stated, that the application of the USLE outside the US may lead to over or under-prediction of actual soil loss. This statement implies that the application of the USLE in the US leads to correct prediction of soil loss. This is not true. Over or under-prediction of actual soil loss rates is also due to the simplicity of the approach. Furthermore, it is stated that the USLE also may lead to uncertainties in predicted soil loss if it is applied to larger scales than the plot scale. Again, this statement implies that predictions for the plot scale are correct, which is not true.

- Page 21, line 26 – 29: redundant to Chapter 2.3

**Technical corrections:**

- Page 2, line 5: "This" → "The"

- Page 5, line 4 - 7: check wording

- Page 6, line 8: "differents" estimates → "different"

- Page 13, line 14: include "the" before "upslope"

- Page 17, line 5: "were" → where

- Page 21, line 14: Grovers → Govers

- Page 21, line 24: include "of" before "watersheds"

- Page 22, line 1 – 2: Check wording / grammar

- Page 22, line 34: available → availability

---

## Referee Comment (RC2) · Anonymous Referee #2 · 17 May 2018

**Review of paper**

**'A review of the (Revised) Universal Soil Loss Equation (R/USLE): with a view to increasing its global applicability and improving soil loss estimates'**

**By R. Benavidez et. al.**

**1. Scope**

The paper provides a thorough introduction into the USLE model family, a group of empirical long term soil erosion models. This paper is of interest to the HESSD community, as the various USLE variants described in this paper are among the most used erosion models overall.

**2. Summary**

The paper gives an introduction into the motivation and method of using USLE models and describes the conceptual background for all individual factors needed to calculate the annual soil loss amounts with USLE models. This is being done by referring to different case studies as well as widely cited papers of variations of USLE models developed to adapt the model to other regions of the world and improve the model family. The calculation formulas of the USLE factors from those papers are provided in tabular form as well, giving a quick overview of these different approaches. The paper also discusses the limitations of USLE models and points at needed future improvements.

**3. General evaluation**

**Scientific significance**

The paper provides a good overview of the topic and goes in depth into the history and motivation of the various USLE models. This is especially helpful for someone just starting with soil erosion modelling. Although mentioned briefly, it is missing a contextualization of USLE models versus other soil erosion modelling approaches.

**Scientific quality**

While providing a useful overview over widely used USLE models and their respective equations as well as discussing the limitations, it could do a better service of evaluating each of the different approaches as well as USLE models performances in general. What is completely missing is any form of information regarding a validation of model results with measurements. Also the connection to surface runoff and sediment transport is missing completely, a very important part of the whole soil erosion process chain and an obvious weak point of the USLE model family. Related to that, the whole sediment delivery ratio (SDR) concept is absent, while being a necessity for most applications of USLE models that go beyond plot scale. Also the paper needs stronger precision and less vagueness in some terms, especially since the target audiences of the paper are newcomers to erosion modelling.

**Presentation quality**

The paper is structured well, but is lacking in visual descriptions of concepts and equations and instead relies too heavily on tabular listing of equations. Especially a visualization of the many (linear and non-linear) equations could make each concept behind it more understandable.

**4. Specific comments**

p. 1, l. 8-10: two minor things, USLE is not necessary the best tool to understand the driving forces behind erosion, due to its dependency on empirical relations and lack of physical based approaches.

Also "effectively manage" is a little presumptuous compared to the little effect some measures actually have when applied (or the little amount of measures that are being enforced in general).

p. 1, l. 23: rather small study cited for such a broad statement. Better or more citations?

p. 2, l. 4-5: "advances in technology" too unspecific.

p. 2, l. 9 + 13: redundant citation.

p. 2, l. 19: average over what precisely, space, time?

p.2, l. 6: contradicting statement regarding sediment transport.

p. 3, l. 10: "things"?! precision please.

p. 3, l. 11: None of the models are being extensively reviewed in this paper, it should be included like the others if this paper is supposed to be providing a complete overview. Also event scale, and the problems with modeling over long-term averages, need to be discussed in regards to the actual processes of erosion.

p. 3, l. 19: As the name suggests ("Universal"), the model in theory was developed for every type of soil, but parameterized for the United States. A noteworthy difference.

p. 3, l. 20: Context of citation should be not in regards to location, but scale.

p. 3, l. 22: first (?) mention of uncertainties with SE models. This needs a more general and honest introduction on its own instead of solely being mentioned at the limitations chapter.

p. 3, l. 22-26: Focus solely on one issue with data (length of data measurements) and is missing more important issues like time step interval length, spatial scale and the amount of variables needed.

p. 5, l. 13-18: noteworthy issue, but should be outside the R-Factor chapter due to its more general nature.

p. 5, l. 23-32: This paragraph reads more like an anecdotal narration of model appliances without any classification or judgement.

p. 5, l. 33-34: This paragraph makes it sound like that's all that's needed to go from annual to monthly time steps, that's a bit misleading.

p. 6, l. 19: Unacceptable figure layout.

p. 11, l. 23-25: How would you test that?

p. 13, l. 20: what is high resolution in this context? Raster cell size is a very important aspect of USLE applications and it's being tip toed around in most papers, so it would be nice to have specific comment to that in this review.

p. 13, l. 27-29: let's be honest, that's the absolute norm in my experience. And that's why raster cell size or use of a proper LS factor calculation is so important and needs to be talked about more critically.

p. 13, l. 30: sounds good, makes sense, but does it improve the model results?

p. 19: very good and short summary of the P-factor, especially with the mention of using it for scenario analysis.

p. 19, l. 13-18: Would be good to comment a bit more on the values from the cited studies from table 10 in this paragraph as well.

p. 20, l. 1: Is there a citable metric behind the citation amount, or is this the expression of a subjective feeling of the author?

p. 20, l. 7: I think this is quite a significant fact which gets ignored most of the time. This should be the actual most common cited limitation...

p. 20, l 11-16: I get the point and it is correct, but I think it is misleading to divert the uncertainties of the USLE modelling results to the data quality or availability, when it is the biggest reason to use the USLE in the first place, over more sophisticated models. Most uncertainties of the USLE stem from the big division between the model design and the actual processes, even when using high resolution data.

p. 20, l 17+: this is such an important paragraph, it should almost be part of the introduction.

p. 21, l. 24: Grammar.

p. 23, l. 15: very true and should honestly be said much earlier in my opinion.

p. 24, l. 2: while the whole paragraph makes a good point, the mention of those conversion factors seems oddly specific at this section.

**5.  Additional comments**

While out of scope for a literature review paper, it would have been very interesting to see the actual soil loss results from each of the presented models compared in a real world or virtual example. It would be quite eye opening, especially for newcomers to erosion modelling, to see the huge variations of results between some models and compared to measurements.

---

## Author Comment (AC1) · 11 Jun 2018

Author's response comments in red.

**AUTHOR'S RESPONSE**

**Referee Comment #1**

**General comments:**

This review paper presents a comprehensive overview of studies applying the R/USLE all over the world and provides information on how different studies have adapted the equations to calculate the factors of the USLE to local conditions. In addition, studies dealing with limitations of the USLE and future developments of the approach are mentioned. The authors explain that they provided this review to serve as a reference for other researchers working with the USLE.

In general, a review of the USLE is well placed in HESS. The authors have done a very diligent work by summarizing many publications applying the USLE. In addition the manuscript provides some helpful hints, as for example the advice to be careful with the units of the USLE-factors used in different studies (i.e. for the K-factor in Chapter 2.2). However, my major objection is, that the manuscript provides only an overview of existing studies and that a critical examination of the approaches presented in the manuscript is missing. Thus, I cannot see a significant own contribution of the authors besides the summary of existing studies on the application of the USLE. The manuscript should thus be thoroughly revised and provide a critical analysis of the approaches presented to gain new insight in the topic. Further comments for revision of the manuscript are given in the following:

- The introduction is very general. It should be worked out, why this review of the USLE is necessary and what is its benefit in relation to other reviews. In addition, the objectives are not clear and included at various locations in the introduction. Thus, the introduction should clearly motivate this review, leading to focused objectives at the end of the introduction (see also specific comments).

Response #1: Previous reviews of soil erosion models were discussed and included a brief mention of the USLE and RUSLE as it has been integrated into other models (page 2, line 8 to 11; page 3: line 2 to 5), but there has not previously been a comprehensive review focussed specifically on covering the RUSLE and all of its components. A review of the related Modified Universal Soil Loss Equation (MUSLE) has been published previously by Sadeghi et al. (2014), and a review of rainfall erosivity has also been done by Nearing et al. (2017). The scope of this paper is to review the entirety of the R/USLE and its all sub-factors and provide a starting point for newer soil erosion modellers to get a handle on the R/USLE depending on their location and data availability, which has not been published previously.

To improve the manuscript, we propose to make the significance and objectives of the review clearer at the beginning of paper, with more critical analysis added in the next iteration of the paper given the comments of referee #1 and referee #2.

- The authors promise, that they will provide guidance which equation is most appropriate for a range of different geoclimatic regions (Page 2, line 17 – 18). However, the advices are very general and the studies presented in Chapter 2 seem to be randomly picked. For example, Chapter 2.1 provides a comprehensive overview of 19 studies that have derived approaches to estimate R-factors for different regions or have applied these approaches (Table 3). Furthermore, the authors summarize various studies using approaches to calculate R-factors in regions other than those for which they were developed. Following this, a simple calculation example is provided (Page 6, line 8 – 16 and Figure 1): In this example, 2

equations developed for Portugal and 1 equation developed for New Zealand are applied to a watershed in New Zealand (Figure 1). As expected, the equations developed for Portugal do not match the seasonal variation in New Zealand. The authors conclude that it is important to understand the regional applicability of rainfall erosivity equations (Page 6, line 17-18). Although many studies were reviewed, the main result of Chapter 2.1 is a very general statement drawn on the basis of a simple example. If such examples are provided, they should cover a much larger number of approaches and data of different regions to derive useful conclusions to guide other users of the USLE. It would be much more important to analyze, if approaches for R-factors could be transferred to regions with similar climate characteristics for which no detailed data is available and what criteria should be applied to do this.

Response #2: Portions of Section 2.1 discuss which datasets and equations are appropriate for locations with annual, monthly, daily, and sub-daily rainfall data (Page 5, line 8+). The studies in Table 3 were chosen due to their scope (global, regional, national) and the fact that their equations had been cited by several other studies in different regions (e.g. the equation by El-Swaify et al. (1987) originally developed in Thailand but also applied in the Philippines and Sri Lanka). Additionally, some equations were chosen because of their utility in predicting intra-annual soil erosion rates (Shamshad et al., 2008; Irvem et al., 2007; Ferreira and Panagopolous, 2014). Page 5 line 33 to page 6 line 7 discusses why estimating seasonal erosion rates is important, especially for areas with high temporal variability of rainfall.

The warning of regional applicability is due to R/USLE studies commonly pointing to rainfall erosivity equations derived in different regions but not justifying why those equations were chosen for their study area. The purpose of testing the different R-factors is to illustrate how the derived rainfall erosivity using the same input data can vary and encourages future users of R/USLE to do the same sensitivity testing in their area. To improve the manuscript, this point will be made clearer and the importance of sensitivity testing will be outlined.

The example from New Zealand is from a more general case study that will form a paper in the future, and we will include a little more guidance from other outcomes, along with summary outcomes from an application in the Philippines in the edited manuscript. In the Philippine case study, the sensitivity testing of the R-factors produced values that were significantly different from each other even though most of the equations were produced near to the Philippines.

To further improve the manuscript, we also propose adding a summary paragraph at the end of each section (rainfall, soil, etc) to critically discuss which datasets and equations are appropriate for general climate types such as hot arid areas, cold arid areas, tropical, and temperate.

- In Chapter 2.2 only studies for the US are presented. It would be interesting, how studies in other regions deal with K-factors?

Response #3: Very true, most studies outside the US use the K-factor equations in Table 4. As mentioned before, a follow-up paper includes a discussion of a New Zealand case study and includes some values from a previous NZ study for K-factor but no equation associated with it (e.g. has a value for loam, clay loam, etc). To improve this review's manuscript, this NZ application and K-factor example will be included.

- Chapter 3 is about limitations of the R/USLE. As before, only existing studies dealing with the limitations of the USLE are summarized and a critical analysis of the limitations is missing (see also specific comments). The topic of validation of estimated soil loss rates by using the

USLE is mentioned only briefly. In my opinion, it is one of the major limitations of the USLE that it is so difficult to validate the estimated soil loss rates. This topic should be discussed in more detail.

Response #4: Referee #2 also made a similar comment about how the uncertainty associated with soil erosion models and USLE is a big limitation and should be spelled out earlier in the paper, possibly the introduction. To improve the manuscript, the uncertainty and lack of validation data limitation of USLE will be mentioned in the introduction and then more critical discussion will be in Section 3. Possible proxies for soil erosion measurements will be mentioned (e.g. water quality data, total suspended sediment loads, comparison to soil erosion rates of similar land cover, etc.) and the paper will also point to global/regional/national studies that have published their soil erosion rates so that future modellers can compare their results with those studies.

- In Chapter 4 again only studies are summarized which are dealing with further developments of the USLE, but again, a critical analysis is missing.

Response #5: The follow-up paper mentioned before discusses the inclusion of other techniques to estimate gully erosion and mass wasting, and that discussion will be incorporated into this review paper instead. The discussion covers the Compound Topographic Index (CTI) for gully erosion, the advantages/disadvantages to using it, and possible ways it can be combined with the RUSLE. Summary critical analysis of this and other recommended further developments (monthly or seasonal erosion and consistency in units) will be added.

- The conclusions are very general.
- The abstract is very brief. It should be thoroughly revised according to the revision of the manuscript.

Response #6: We propose to include more critical analysis and results in the revised conclusions and abstracts.

**Specific comments:**

Response #7: In general, most of these specific comments will be incorporated in the next iteration of this paper.

- Page 1, line 22 – Page 2, line 5: The introductory part on soil erosion is very long and not specific for the USLE. It should be shortened and focused.

Response #8: Noted.

- Page 2, line 6 – 13: In this section, a few review papers on erosion models are presented. It is not clear, why these reviews have been selected. I suggest to focus on previous reviews of the USLE and to work out, why the additional review presented in this paper is necessary and what will be the benefit of it.

Response #8: As previously discussed in Response #1, previous soil erosion reviews have covered soil erosion models in general and have mentioned USLE, but not discussed it in depth. One review paper of rainfall erosivity has been previously published, but there are no published reviews focusing only on the R/USLE, its components, and previous applications.

- Page 2, line 15 – 19: I suggest to move this section to the objectives at the end of the introduction.

Response #9: Noted, will be included as part of objectives.

- Page 2, line 28 – 29: move to objectives at the end of the introduction. In addition, it should be made clear, which limitations of the USLE are analyzed.

Response #10: Noted, will be included as part of objectives.

- Page 2, line 30 – 34: redundant to the section above. Include the information not yet provided in line 19 – 27 into this section.

Response #11: Noted.

- Page 3, line 10 – 13: The objectives of the study mentioned at various locations in the introduction should be summarized at the end of the introduction (see comments above).

Response #12: Noted, and agreed that these repeated points will be summarised at the end of the introduction section as clear objectives.

- Page 3, line 19 – 26: In my opinion, this information fits better in the introduction.

Response #13: Noted, and will be incorporated into the introduction.

- Page 4, line 1 – 5: some additional objectives are mentioned in this section → should be moved to a focused section presenting the objectives at the end of the introduction.

Response #14: Pursuant to previous comments, these will be incorporated into the objectives section.

- Page 3, Chapter 2: Some general information on the USLE should be provided, i.e. that it was developed from soil loss rates on plot experiments.

Response #15: Noted, we propose including some general information about how USLE was formulated, including mention of the unit plot.

- Page 11, line 6: the information on the R/USLE unit plot is also essential for the other factors. It should be mentioned in the preface of Chapter 2, i.e. page 3, line 19 - 26.

Response #16: Noted, and will be incorporated into Section 2.

- Page 20, line 2 – 10: in this paragraph it is stated, that the application of the USLE outside the US may lead to over or under-prediction of actual soil loss. This statement implies that the application of the USLE in the US leads to correct prediction of soil loss. This is not true. Over or under-prediction of actual soil loss rates is also due to the simplicity of the approach. Furthermore, it is stated that the USLE also may lead to uncertainties in predicted soil loss if it is applied to larger scales than the plot scale. Again, this statement implies that predictions for the plot scale are correct, which is not true.

Response #17: Agreed, and the wording of this section will be changed to make it more clear that the uncertainties associated with USLE are not just dependent on the study site application but also on the simplified approach vs the complex interactions associated with soil loss.

- Page 21, line 26 – 29: redundant to Chapter 2.3

Response #18: All technical corrections are noted and will be changed in the next iteration of this paper.

**Referee Comment #2**

**1. Scope**

The paper provides a thorough introduction into the USLE model family, a group of empirical long term soil erosion models. This paper is of interest to the HESSD community, as the various USLE variants described in this paper are among the most used erosion models overall.

**2. Summary**

The paper gives an introduction into the motivation and method of using USLE models and describes the conceptual background for all individual factors needed to calculate the annual soil loss amounts with USLE models. This is being done by referring to different case studies as well as widely cited papers of variations of USLE models developed to adapt the model to other regions of the world and improve the model family. The calculation formulas of the USLE factors from those papers are provided in tabular form as well, giving a quick overview of these different approaches. The paper also discusses the limitations of USLE models and points at needed future improvements.

**3. General evaluation**

**Scientific significance**

The paper provides a good overview of the topic and goes in depth into the history and motivation of the various USLE models. This is especially helpful for someone just starting with soil erosion modelling. Although mentioned briefly, it is missing a contextualization of USLE models versus other soil erosion modelling approaches.

Response #19: Please see Response #1 regarding the place of USLE in other soil erosion models and reviews. To improve the manuscript, we propose emphasising the place of USLE within the soil erosion modelling space will be emphasised and the reader will be directed to more general erosion reviews.

**Scientific quality**

While providing a useful overview over widely used USLE models and their respective equations as well as discussing the limitations, it could do a better service of evaluating each of the different approaches as well as USLE models performances in general. What is completely missing is any form of information regarding a validation of model results with measurements. Also the connection to surface runoff and sediment transport is missing completely, a very important part of the whole soil erosion process chain and an obvious weak point of the USLE model family. Related to that, the whole sediment delivery ratio (SDR) concept is absent, while being a necessity for most applications of USLE models that go beyond plot scale. Also the paper needs stronger precision and less vagueness in some terms, especially since the target audiences of the paper are newcomers to erosion modelling.

Response #20: Please see Response #4 regarding validation of soil erosion results using proxies. We propose to add a discussion about the importance of data validation, how sediment data collection is expensive, therefore there is a need to compile global and national databases of sediment data/soil erosion measurements, which is a good point for future work. SDR was mentioned in some of the papers that were cited in this review, and will be now be included as a discussion point in Section 4.1 about representing other types of erosion, and possibly in the LS-factor discussion instead some LS-factor approaches use flow accumulation.

**Presentation quality**

The paper is structured well, but is lacking in visual descriptions of concepts and equations and instead relies too heavily on tabular listing of equations. Especially a visualization of the many (linear and non-linear) equations could make each concept behind it more understandable.

Response #21: Only a few of the cited equations have published graphs of their equations, and the next iteration can include some of these graphs lifted from their paper with proper citation. Some maps of the output will also be included to show how the different equations produce different sub-factors that affected the soil loss estimates.

**4. Specific comments**

Response #22: These specific comments will be incorporated in the next iteration of the paper as they are very constructive. Issues around wording require more clarification and precision from the authors. More critical analysis will be added.

p. 1, l. 8-10: two minor things, USLE is not necessary the best tool to understand the driving forces behind erosion, due to its dependency on empirical relations and lack of physical based approaches. Also "effectively manage" is a little presumptuous compared to the little effect some measures actually have when applied (or the little amount of measures that are being enforced in general).

Response #23: True, although RUSLE modelling can give management an idea of what kind of management interventions prevent soil erosion (e.g. bare soil vs contouring vs mulching). In the more general case study paper, scenario analysis was done for the Philippines case study showing decreases in potential soil loss when conservation technologies were applied in agricultural areas. We propose adding some of these output maps showing the difference in soil loss due to the conservation technologies.

p. 1, l. 23: rather small study cited for such a broad statement. Better or more citations?

Response #24: Noted, will add further citations.

p. 2, l. 4-5: "advances in technology" too unspecific.

Response #25: Will be more specific (e.g. GIS programmes for spatial analysis, increases in desktop computing power, etc) in next iteration of paper.

p. 2, l. 9 + 13: redundant citation.

Response #26: Noted.

p. 2, l. 19: average over what precisely, space, time?

Response #27: Space and time as the soil loss is in estimates of tons hectare$^{-1}$ year$^{-1}$

p.2, l. 6: contradicting statement regarding sediment transport.

Response #28: The statement reads "Soil erosion models aid land management by helping understand sediment transport and its effects on a landscape". The model outputs help elucidate driving forces/possible causes of soil erosion, sediment transport, and the potential degrading effects on landscape. We are unclear as to where the contradiction is in this statement, and would appreciate further clarity from the Reviewer.

p. 3, l. 10: "things"?! precision please.

Response #29: "Things" refers to choices in sub-equations, caveats associated with RUSLE, limitations, etc. We propose clarifying this in the manuscript by replacing "things" with "factors such as sub-equations, limitations," etc.

p. 3, l. 11: None of the models are being extensively reviewed in this paper, it should be included like the others if this paper is supposed to be providing a complete overview. Also event scale, and the problems with modeling over long-term averages, need to be discussed in regards to the actual processes of erosion.

Response #30: This review mainly focuses on USLE and RUSLE, since the event-based MUSLE has already been extensively reviewed by Sadeghi et al. (2014). To improve the manuscript, some of the issues associated with modelling over long-term averages and event-based erosion events will be discussed.

p. 3, l. 19: As the name suggests ("Universal"), the model in theory was developed for every type of soil, but parameterized for the United States. A noteworthy difference.

Response #31: True, and I verbally made a point about this in an oral presentation in December 2018 entitled "Parameterisation of the Revised Universal Soil Loss Equation (RUSLE) for New Zealand Data and Conditions" to discuss the regional limitations of the RUSLE. Will be incorporated into this chapter because it is an important limitation.

p. 3, l. 20: Context of citation should be not in regards to location, but scale.

Response #32: Sentence will be reworded for clarity.

p. 3, l. 22: first (?) mention of uncertainties with SE models. This needs a more general and honest introduction on its own instead of solely being mentioned at the limitations chapter.

Response #33: This issue has been raised by RC#1 and the uncertainty of soil erosion models will be emphasised and placed in the introduction since this limitation came up many times in the papers reviewed.

p. 3, l. 22-26: Focus solely on one issue with data (length of data measurements) and is missing more important issues like time step interval length, spatial scale and the amount of variables needed.

Response #34: Although these issues are outlined in each of the factor sections, these lines will be expanded to include those other issues.

p. 5, l. 13-18: noteworthy issue, but should be outside the R-Factor chapter due to its more general nature.

Response #35: Inconsistencies in units is brought up later on in the limitations section.

p. 5, l. 23-32: This paragraph reads more like an anecdotal narration of model appliances without any classification or judgement.

Response #36: Section contextualises that monthly rainfall records can be used instead of storm records that were in the original USLE.

p. 5, l. 33-34: This paragraph makes it sound like that's all that's needed to go from annual to monthly time steps, that's a bit misleading.

Response #37: The R-factor equations that estimate monthly erosivity to calculate annual erosivity have been used by some RUSLE applications to estimate monthly/seasonal soil loss by only varying the R-factor. This will be clarified and the paper will point to those studies.

p. 6, l. 19: Unacceptable figure layout.

Response #38: Unsure what this means, please clarify.

p. 11, l. 23-25: How would you test that?

Response #39: Similar to the sensitivity analysis of the R-factor equations, testing the K-factor equations to see which ones produce values similar to each other or significantly different from the others could be one way of testing their applicability. Another way would be to compare the derived K-factor values with published values from similar soils.

p. 13, l. 20: what is high resolution in this context? Raster cell size is a very important aspect of USLE applications and it's being tip toed around in most papers, so it would be nice to have specific comment to that in this review.

Response #40: High resolution will vary depending on scale, but Panagos et al. (2015a) talked about 100m resolution DEMs having an associated loss of detail regarding flow network compared to 25m resolution DEMs.

p. 13, l. 27-29: let's be honest, that's the absolute norm in my experience. And that's why raster cell size or use of a proper LS factor calculation is so important and needs to be talked about more critically.

Response #41: Agreed, and it was touched on briefly but will be further clarified here.

p. 13, l. 30: sounds good, makes sense, but does it improve the model results?

Response #42: As mentioned in previous comments, there is a follow-up paper to this one and it includes sensitivity testing for LS-factor using the method that only accounts for slope and length against a method that incorporates flow accumulation. It was found that with high resolution DEMS (15m and finer), the first method was better at the watershed scale for delineating large areas that can be marked for soil conservation measures while the second method would be better at the sub-watershed or field scale. Those results will be briefly mentioned in this section in the next iteration of the paper.

p. 19: very good and short summary of the P-factor, especially with the mention of using it for scenario analysis.

Response #43: Thank you.

p. 19, l. 13-18: Would be good to comment a bit more on the values from the cited studies from table 10 in this paragraph as well.

Response #44: Noted, will be more clear about these values and their possible effect on soil loss estimates.

p. 20, l. 1: Is there a citable metric behind the citation amount, or is this the expression of a subjective feeling of the author?

Response #45: This limitation came up in most, if not all, of the studies that were reviewed that applied the R/USLE to an area outside of the USA. To clarify, we propose citing a few of the studies and reviews that discussed the limitations of applying RUSLE outside the USA.

p. 20, l. 7: I think this is quite a significant fact which gets ignored most of the time. This should be the actual most common cited limitation…

Response #46: True, and as per the comments of Referee #1, the unit plot will be emphasised in the introduction of the RUSLE equation.

p. 20, l 11-16: I get the point and it is correct, but I think it is misleading to divert the uncertainties of the USLE modelling results to the data quality or availability, when it is the biggest reason to use the USLE in the first place, over more sophisticated models. Most uncertainties of the USLE stem from the big division between the model design and the actual processes, even when using high-resolution data.

Response #47: True, and will reword and add more critical analysis.

p. 20, l 17+: this is such an important paragraph, it should almost be part of the introduction.

Response #48: True, and will be mentioned in the introduction.

p. 21, l. 24: Grammar.

Response #49: Sentence will be revised.

p. 23, l. 15: very true and should honestly be said much earlier in my opinion.

Response #50: Noted, will be brought up earlier.

p. 24, l. 2: while the whole paragraph makes a good point, the mention of those conversion factors seems oddly specific at this section.

Response #51: This sentence was meant to reiterate making sure that units were consistent, another summary sentence will be written for this section.

**5. Additional comments**

While out of scope for a literature review paper, it would have been very interesting to see the actual soil loss results from each of the presented models compared in a real world or virtual example. It would be quite eye opening, especially for newcomers to erosion modelling, to see the huge variations of results between some models and compared to measurements.

Response #52: This is the scope of the follow-up paper that applies the RUSLE to New Zealand and Philippines study areas, including sensitivity testing and comparison to measured data.

---

## Author Response (AR1)

**AUTHOR'S RESPONSE #2 (August 2 2018)**

Author's response comments in red.

**Editor Decision Comments**

- A critical discussion/evaluation of the R/USLE applications presented in the paper. Have they been applied according to the recommendations, are output uncertainties honestly discussed, etc.

The applications presented, along with their recommendations and outcomes, are briefly discussed throughout the paper with the relevant information placed within each sub-factor section. Discussion of these applications was added to the limitations section which has been completely overhauled to include more critical discussion about previous R/USLE applications and the uncertainties associated with the model, the data, and the results.

- R/USLE should be put into the greater context of available Soil Loss Models (which models are out there, how do they differ with respect to underlying concepts/simplifications/assumptions, when to use which, data requirements and output quality)

There have already been previously published extensive reviews of the various soil loss models (Aksoy & Kavvas, 2005; de Vente & Poesen, 2005; Merritt et al., 2003) that discuss the R/USLE and its place in the soil erosion modelling sphere, as well as the concepts/data/quality of other soil loss models. This review briefly discusses the R/USLE within the modelling sphere but since the soil loss model reviews already exist, the reader is referred to those papers instead.

- A critical and honest discussion of R/USLE output quality / predictive uncertainty through comparison with observations and/or other models. I know that related observations are sparse, but this makes collection and comparison even more valuable for a review paper.

The overhauled limitations section more critically discusses the uncertainties associated with R/USLE and other soil erosion models, the importance of observational data, proxies to validate the model results, and recommendations to assemble a database of soil loss values is discussed in the limitations section to address uncertainty. This section also discusses the RUSLE applications that did compare their model results to observational data, or to other models.

- Recommendations on the application of R/USLE, not only but also with respect to spatial resolution (minimum resolution, recommended resolution, effect of resolution on uncertainty)

At the end of each sub-factor section, a summary and critical analysis paragraph has been added to more clearly recommend equations for R/USLE applications. The issue of spatial resolution is brought up more clearly in the revised LS-factor section.

- P2 / L6: The key issue here is the word 'transport': Many soil loss models do not explicitly include transport simulations, they just provide erosion rates. So you may want to replace 'by helping understand sediment transport ..' with 'by helping predict erosion rates'

Noted and changed.

- P6 / L19: The values for summer, winter etc. are independent, so why connect the values from the different studies with lines? Either show as multi-bar plot, or display in a table only.

Noted and changed to show the soil loss values as multi-bar plot and the rainfall as lines instead.

**Referee Comment #1**

**General comments:**

This review paper presents a comprehensive overview of studies applying the R/USLE all over the world and provides information on how different studies have adapted the equations to calculate the factors of the USLE to local conditions. In addition, studies dealing with limitations of the USLE and future developments of the approach are mentioned. The authors explain that they provided this review to serve as a reference for other researchers working with the USLE.

In general, a review of the USLE is well placed in HESS. The authors have done a very diligent work by summarizing many publications applying the USLE. In addition the manuscript provides some helpful hints, as for example the advice to be careful with the units of the USLE-factors used in different studies (i.e. for the K-factor in Chapter 2.2). However, my major objection is, that the manuscript provides only an overview of existing studies and that a critical examination of the approaches presented in the manuscript is missing. Thus, I cannot see a significant own contribution of the authors besides the summary of existing studies on the application of the USLE. The manuscript should thus be thoroughly revised and provide a critical analysis of the approaches presented to gain new insight in the topic. Further comments for revision of the manuscript are given in the following:

- The introduction is very general. It should be worked out, why this review of the USLE is necessary and what is its benefit in relation to other reviews. In addition, the objectives are not clear and included at various locations in the introduction. Thus, the introduction should clearly motivate this review, leading to focused objectives at the end of the introduction (see also specific comments).

Previous reviews of soil erosion models were discussed and included a brief mention of the USLE and RUSLE as it has been integrated into other models, but there has not previously been a comprehensive review focussed specifically on covering the RUSLE and all of its components. A review of the related Modified Universal Soil Loss Equation (MUSLE) has been published previously by Sadeghi et al. (2014), and a review of rainfall erosivity has also been done by Nearing et al. (2017). The scope of this paper is to review the entirety of the R/USLE and its all sub-factors and provide a starting point for newer soil erosion modellers to get a handle on the R/USLE depending on their location and data availability, which has not been published previously.

The objectives section has been included in this iteration of the paper to make the significance of this review more clear (P3, L25+).

- The authors promise, that they will provide guidance which equation is most appropriate for a range of different geoclimatic regions (Page 2, line 17 – 18). However, the advices are very general and the studies presented in Chapter 2 seem to be randomly picked. For example, Chapter 2.1 provides a comprehensive overview of 19 studies that have derived approaches to estimate R-factors for different regions or have applied these approaches (Table 3). Furthermore, the authors summarize various studies using approaches to calculate R-factors in regions other than those for which they were developed. Following this, a simple calculation example is provided (Page 6, line 8 – 16 and Figure 1): In this example, 2 equations developed for Portugal and 1 equation developed for New Zealand are applied to a watershed in New Zealand (Figure 1). As expected, the equations developed for Portugal do not match the seasonal variation in New Zealand. The authors conclude that it is

important to understand the regional applicability of rainfall erosivity equations (Page 6, line 17-18). Although many studies were reviewed, the main result of Chapter 2.1 is a very general statement drawn on the basis of a simple example. If such examples are provided, they should cover a much larger number of approaches and data of different regions to derive useful conclusions to guide other users of the USLE. It would be much more important to analyze, if approaches for R-factors could be transferred to regions with similar climate characteristics for which no detailed data is available and what criteria should be applied to do this.

Portions of Section 2.1 discuss which datasets and equations are appropriate for locations with annual, monthly, daily, and sub-daily rainfall data. The studies in Table 3 were chosen due to their scope (global, regional, national) and the fact that their equations had been cited by several other studies in different regions (e.g. the equation by El-Swaify et al. (1987) originally developed in Thailand but also applied in the Philippines and Sri Lanka). Additionally, some equations were chosen because of their utility in predicting intra-annual soil erosion rates (Shamshad et al., 2008; Irvem et al., 2007; Ferreira and Panagopolous, 2014). P6 L29+ discusses why estimating seasonal erosion rates is important, especially for areas with high temporal variability of rainfall.

The warning of regional applicability is due to R/USLE studies commonly pointing to rainfall erosivity equations derived in different regions but not justifying why those equations were chosen for their study area. The purpose of testing the different R-factors is to illustrate how the derived rainfall erosivity using the same input data can vary and encourages future users of R/USLE to do the same sensitivity testing in their area. The importance of sensitivity testing was outlined more clearly in this iteration of the paper, mainly in the New Zealand example on P7 L6 to L18.

To improve the manuscript, a summary paragraph was added at the end of each section (rainfall, soil, etc) to critically discuss which datasets and equations are appropriate for different climate types, spatial resolution, temporal scales, data availability etc.

- In Chapter 2.2 only studies for the US are presented. It would be interesting, how studies in other regions deal with K-factors?

Response #3: Very true, most studies outside the US use the K-factor equations in Table 4. As mentioned before, a follow-up paper includes a discussion of a New Zealand case study and includes some values from a previous NZ study for K-factor but no equation associated with it (e.g. has a value for loam, clay loam, etc). To improve this manuscript, more examples of K-factors from New Zealand and the Philippines were added.

- Chapter 3 is about limitations of the R/USLE. As before, only existing studies dealing with the limitations of the USLE are summarized and a critical analysis of the limitations is missing (see also specific comments). The topic of validation of estimated soil loss rates by using the USLE is mentioned only briefly. In my opinion, it is one of the major limitations of the USLE that it is so difficult to validate the estimated soil loss rates. This topic should be discussed in more detail.

Referee #2 also made a similar comment about how the uncertainty associated with soil erosion models and USLE is a big limitation and should be spelled out earlier in the paper, possibly the introduction. To improve the manuscript, the uncertainty and lack of validation data limitation of USLE was mentioned in the introduction (P2 L18 to L21) and then more critical discussion in Section 3. The limitations section discusses the uncertainties in depth as well as possible proxies for soil

erosion measurements (e.g. water quality data, total suspended sediment loads, comparison to soil erosion rates of similar land cover, etc.).

- In Chapter 4 again only studies are summarized which are dealing with further developments of the USLE, but again, a critical analysis is missing.

The follow-up paper mentioned before discusses the inclusion of other techniques to estimate gully erosion and mass wasting, and that discussion will be incorporated into this review paper instead. The discussion covers the Compound Topographic Index (CTI) for gully erosion, the advantages/disadvantages to using it, and possible ways it can be combined with the RUSLE. This section has also been changed to discuss more critically why these further developments are needed.

- The conclusions are very general.
- The abstract is very brief. It should be thoroughly revised according to the revision of the manuscript.

Both the conclusions and abstract have been revised to reflect the more detailed and critical version of the manuscript.

**Specific comments:**

In general, most of these specific comments were incorporated in the next iteration of this paper.

- Page 1, line 22 – Page 2, line 5: The introductory part on soil erosion is very long and not specific for the USLE. It should be shortened and focused.

This part is needed to establish background and context of why soil erosion is a problem, and why soil erosion models in general are needed.

- Page 2, line 6 – 13: In this section, a few review papers on erosion models are presented. It is not clear, why these reviews have been selected. I suggest to focus on previous reviews of the USLE and to work out, why the additional review presented in this paper is necessary and what will be the benefit of it.

As previously discussed, previous soil erosion reviews have covered soil erosion models in general and have mentioned USLE, but not discussed it in depth. One review paper of rainfall erosivity has been previously published, but there are no published reviews focusing only on the R/USLE, its components, and previous applications. Added clarification regarding the scope of the other soil erosion reviews, and established that this paper focuses specifically on the R/USLE.

- Page 2, line 15 – 19: I suggest to move this section to the objectives at the end of the introduction.

This part is now in the objectives section.

- Page 2, line 28 – 29: move to objectives at the end of the introduction. In addition, it should be made clear, which limitations of the USLE are analyzed.

This part is now in the objectives section.

- Page 2, line 30 – 34: redundant to the section above. Include the information not yet provided in line 19 – 27 into this section.

Revised.

- Page 3, line 10 – 13: The objectives of the study mentioned at various locations in the introduction should be summarized at the end of the introduction (see comments above).

Added the objectives section.

- Page 3, line 19 – 26: In my opinion, this information fits better in the introduction.

The original conditions under which the R/USLE was formulated are now included in the introduction (P2 L30).

- Page 4, line 1 – 5: some additional objectives are mentioned in this section → should be moved to a focused section presenting the objectives at the end of the introduction.

Pursuant to previous comments, these were incorporated into the objectives section.

- Page 3, Chapter 2: Some general information on the USLE should be provided, i.e. that it was developed from soil loss rates on plot experiments.

Discussion about the unit plot was included (P4 L12 to L13).

- Page 11, line 6: the information on the R/USLE unit plot is also essential for the other factors. It should be mentioned in the preface of Chapter 2, i.e. page 3, line 19 - 26.

The unit plot and original conditions under which the R/USLE was formulated are now included in the information about the USLE (P4 L6 to L12).

- Page 20, line 2 – 10: in this paragraph it is stated, that the application of the USLE outside the US may lead to over or under-prediction of actual soil loss. This statement implies that the application of the USLE in the US leads to correct prediction of soil loss. This is not true. Over or under-prediction of actual soil loss rates is also due to the simplicity of the approach. Furthermore, it is stated that the USLE also may lead to uncertainties in predicted soil loss if it is applied to larger scales than the plot scale. Again, this statement implies that predictions for the plot scale are correct, which is not true.

The wording of this section was changed to make it more clear that the uncertainties associated with USLE are not just dependent on the study site application but also on the simplified approach vs the complex interactions associated with soil loss.

- Page 21, line 26 – 29: redundant to Chapter 2.3

All technical corrections are noted and will be changed in the next iteration of this paper.

Referee Comment #2

**1. Scope**

The paper provides a thorough introduction into the USLE model family, a group of empirical long term soil erosion models. This paper is of interest to the HESSD community, as the various USLE variants described in this paper are among the most used erosion models overall.

**2. Summary**

The paper gives an introduction into the motivation and method of using USLE models and describes the conceptual background for all individual factors needed to calculate the annual soil loss amounts with USLE models. This is being done by referring to different case studies as well as widely cited papers of variations of USLE models developed to adapt the model to other regions of the world and

improve the model family. The calculation formulas of the USLE factors from those papers are provided in tabular form as well, giving a quick overview of these different approaches. The paper also discusses the limitations of USLE models and points at needed future improvements.

**3. General evaluation**

**Scientific significance**

The paper provides a good overview of the topic and goes in depth into the history and motivation of the various USLE models. This is especially helpful for someone just starting with soil erosion modelling. Although mentioned briefly, it is missing a contextualization of USLE models versus other soil erosion modelling approaches.

To improve the manuscript, the place of USLE within the soil erosion modelling space was emphasised (P3 L7 to 14) and the reader is directed to more general erosion reviews (P3 L15 to L19).

**Scientific quality**

While providing a useful overview over widely used USLE models and their respective equations as well as discussing the limitations, it could do a better service of evaluating each of the different approaches as well as USLE models performances in general. What is completely missing is any form of information regarding a validation of model results with measurements. Also the connection to surface runoff and sediment transport is missing completely, a very important part of the whole soil erosion process chain and an obvious weak point of the USLE model family. Related to that, the whole sediment delivery ratio (SDR) concept is absent, while being a necessity for most applications of USLE models that go beyond plot scale. Also the paper needs stronger precision and less vagueness in some terms, especially since the target audiences of the paper are newcomers to erosion modelling.

To address this, the limitations section was overhauled to discuss:

- Uncertainty in the R/USLE and in soil erosion models in general
- Possible proxies to validate soil erosion predictions, such as water quality data
- The SDR
- The need to compile global and national databases of sediment data/soil erosion measurements, which is a good point for future work.

**Presentation quality**

The paper is structured well, but is lacking in visual descriptions of concepts and equations and instead relies too heavily on tabular listing of equations. Especially a visualization of the many (linear and non-linear) equations could make each concept behind it more understandable.

Only a few of the cited studies have published graphs of their equations, and although it was suggested in earlier response to include maps of the output, those output maps may fit better in a case study paper instead of a review paper.

**4. Specific comments**

Most of these specific comments were incorporated in the next iteration of the paper as they are very constructive. Issues around wording require more clarification and precision from the authors, and was addressed in the revised manuscript.

p. 1, l. 8-10: two minor things, USLE is not necessary the best tool to understand the driving forces behind erosion, due to its dependency on empirical relations and lack of physical based approaches. Also "effectively manage" is a little presumptuous compared to the little effect some measures actually have when applied (or the little amount of measures that are being enforced in general).

True, although RUSLE modelling can give management an idea of what kind of management interventions prevent soil erosion (e.g. bare soil vs contouring vs mulching). The P-factor section does briefly mention the results of a case study where scenario analysis was done for the Philippines case study showing decreases in potential soil loss when conservation technologies were applied in agricultural areas (P12 L20 to L24).

p. 1, l. 23: rather small study cited for such a broad statement. Better or more citations?

More citations added.

p. 2, l. 4-5: "advances in technology" too unspecific.

The more specific attributes (e.g. GIS programmes for spatial analysis, increases in desktop computing power, etc.) were added (P2 L14 to L15).

p. 2, l. 9 + 13: redundant citation.

Noted and removed.

p. 2, l. 19: average over what precisely, space, time?

Space and time as the soil loss is in estimates of tons hectare$^{-1}$ year$^{-1}$, added to clarify.

p.2, l. 6: contradicting statement regarding sediment transport.

Soil erosion models aid land management by elucidating driving forces/possible causes of soil erosion, sediment transport, and the potential degrading effects on landscape. Reworded this section to make it more clear.

p. 3, l. 10: "things"?! precision please.

 "Things" refers to choices in sub-equations, caveats associated with RUSLE, limitations, etc. This was clarified in the manuscript by replacing "things" with "factors such as sub-equations, limitations," etc.

p. 3, l. 11: None of the models are being extensively reviewed in this paper, it should be included like the others if this paper is supposed to be providing a complete overview. Also event scale, and the problems with modeling over long-term averages, need to be discussed in regards to the actual processes of erosion.

This review mainly focuses on USLE and RUSLE, since the event-based MUSLE has already been extensively reviewed by Sadeghi et al. (2014). To improve the manuscript, some of the issues associated with modelling over long-term averages and event-based erosion events and the uncertainties of the R/USLE in general are discussed in the limitations section.

p. 3, l. 19: As the name suggests ("Universal"), the model in theory was developed for every type of soil, but parameterized for the United States. A noteworthy difference.

True, and I verbally made a point about this in an oral presentation in December 2018 entitled "Parameterisation of the Revised Universal Soil Loss Equation (RUSLE) for New Zealand Data and

Conditions" to discuss the regional limitations of the RUSLE. A warning is given (P4 L8 to L11) regarding careful parameterisation outside of the USA.

p. 3, l. 20: Context of citation should be not in regards to location, but scale.

Sentence was reworded for clarity.

p. 3, l. 22: first (?) mention of uncertainties with SE models. This needs a more general and honest introduction on its own instead of solely being mentioned at the limitations chapter.

The uncertainties associated with soil erosion models is now briefly mentioned in the introduction (P2 L19 to L21) but also given a more critical discussion in the overhauled limitations section.

p. 3, l. 22-26: Focus solely on one issue with data (length of data measurements) and is missing more important issues like time step interval length, spatial scale and the amount of variables needed.

These issues are outlined in each of the factor sections with further discussion. To expand on these lines in this section, issues such as timestep and spatial variation were included as examples.

p. 5, l. 13-18: noteworthy issue, but should be outside the R-Factor chapter due to its more general nature.

Inconsistencies in units is brought up later on in the limitations section, and the lack of consistency is also included as a point of future work.

p. 5, l. 23-32: This paragraph reads more like an anecdotal narration of model appliances without any classification or judgement.

Section contextualises that monthly rainfall records can be used instead of storm records that were in the original USLE, and re-establishes why monthly soil erosion estables aer useful.

p. 5, l. 33-34: This paragraph makes it sound like that's all that's needed to go from annual to monthly time steps, that's a bit misleading.

The R-factor equations that estimate monthly erosivity to calculate annual erosivity have been used by some RUSLE applications to estimate monthly/seasonal soil loss by only varying the R-factor. This point has been clarified.

p. 6, l. 19: Unacceptable figure layout.

Pursuant to the comments of the editor, this figure has been revised.

p. 11, l. 23-25: How would you test that?

Similar to the sensitivity analysis of the R-factor equations, testing the K-factor equations to see which ones produce values similar to each other or significantly different from the others could be one way of testing their applicability. Another way would be to compare the derived K-factor values with published values from similar soils. This information has now been added to the manuscript.

p. 13, l. 20: what is high resolution in this context? Raster cell size is a very important aspect of USLE applications and it's being tip toed around in most papers, so it would be nice to have specific comment to that in this review.

High resolution will vary depending on scale, and is also an issue of vertical accuracy in the freely available global and national DEMs. This point is discussed in P10 L4 to L9.

p. 13, l. 27-29: let's be honest, that's the absolute norm in my experience. And that's why raster cell size or use of a proper LS factor calculation is so important and needs to be talked about more critically.

Agreed, and it was touched on briefly but is further clarified here through the summary paragraph of the LS-factor section.

p. 13, l. 30: sounds good, makes sense, but does it improve the model results?

As mentioned in previous comments, there is a follow-up paper to this one and it includes sensitivity testing for LS-factor using the method that only accounts for slope and length against a method that incorporates flow accumulation. It was found that with high resolution DEMS (15m and finer), the first method was better at the watershed scale for delineating large areas that can be marked for soil conservation measures while the second method would be better at the sub-watershed or field scale. Those results are briefly mentioned in this section in the next iteration of the paper.

p. 19: very good and short summary of the P-factor, especially with the mention of using it for scenario analysis.

Thank you.

p. 19, l. 13-18: Would be good to comment a bit more on the values from the cited studies from table 10 in this paragraph as well.

Noted, revised manuscript more clear about these values and their possible effect on soil loss estimates.

p. 20, l. 1: Is there a citable metric behind the citation amount, or is this the expression of a subjective feeling of the author?

This limitation came up in most, if not all, of the studies that were reviewed that applied the R/USLE to an area outside of the USA. To clarify, a few of the studies and reviews that discussed the limitations of applying RUSLE outside the USA are cited.

p. 20, l. 7: I think this is quite a significant fact which gets ignored most of the time. This should be the actual most common cited limitation…

True, and as per the comments of Referee #1, the unit plot is now emphasised in the introduction of the RUSLE equation.

p. 20, l 11-16: I get the point and it is correct, but I think it is misleading to divert the uncertainties of the USLE modelling results to the data quality or availability, when it is the biggest reason to use the USLE in the first place, over more sophisticated models. Most uncertainties of the USLE stem from the big division between the model design and the actual processes, even when using high-resolution data.

True, and these limitations are more critically discussed in the now-overhauled limitations section.

p. 20, l 17+: this is such an important paragraph, it should almost be part of the introduction.

True, and is now mentioned in the introduction.

p. 21, l. 24: Grammar.

Sentence revised.

p. 23, l. 15: very true and should honestly be said much earlier in my opinion.

Noted, and now brought up earlier.

p. 24, l. 2: while the whole paragraph makes a good point, the mention of those conversion factors seems oddly specific at this section.

This sentence was meant to reiterate making sure that units were consistent, another summary sentence was written for this section.

**5. Additional comments**

While out of scope for a literature review paper, it would have been very interesting to see the actual soil loss results from each of the presented models compared in a real world or virtual example. It would be quite eye opening, especially for newcomers to erosion modelling, to see the huge variations of results between some models and compared to measurements.

This is the scope of the follow-up paper that applies the RUSLE to New Zealand and Philippines study areas, including sensitivity testing and comparison to measured data.

[revised manuscript text omitted]

• 1: Very fine granular
• 2: Fine granular
• 3: Medium or coarse granular
• 4: Blocky, platy, or massive
c = Profile-permeability class
• 1: Rapid
• 2: Moderate to rapid
• 3: Moderate
• 4: Slow to moderate
• 5: Slow
• 6: Very slow | Thailand (Eiumnoh, 2000); Vanuatu (Dumas & Fossey, 2009); Philippines (Schmitt, 2009); India (Jain & Das, 2010); Turkey (Ozsoy et al., 2012); Iran (Bagherzadeh, 2014); Portugal (Ferreira & Panagopoulos, 2014); China (Li et al., 2014); European Union (Panagos et al., 2014) |
| 2 | Williams and Renard (1983) as cited in Chen et al. (2011) | USA | Sand (%), silt (%), clay (%), organic carbon (%) | $K = 0.2 + 0.3 \exp\left(0.0256 \times Sa \times \left(1 - \dfrac{Si}{100}\right)\right)$

$\times \left(\dfrac{Si}{Cl + Si}\right)^{0.3}$

$\times \left(1.0 - \dfrac{0.25 \times C}{C + \exp(3.72 - 2.95C)}\right)$

$\times \Big(1.0$
$- \dfrac{0.7 \times SN}{SN + \exp(-5.51 + 22.9SN)}\Big)$

Sa = Sand %
Si = Silt %
Cl = Clay %
SN = 1-(Sa/100)
C = Organic Carbon | China (Chen et al., 2011) |
| 3 | David (1988), a simplified | USA | Sand (%), clay (%), silt (%), organic | $K = [(0.043 \times pH) + (0.62 \div OM) + (0.0082 \times S)$
$- (0.0062 \times C)] \times Si$ | Philippines (David, 1988; |

| | | | version of Wischmeier and Mannering (1969) | matter (%), pH | pH = pH of the soil
 OM = Organic matter in percent
 S = Sand content in percent
 C = Clay ratio = % clay / (% sand + % silt)
 Si = Silt content = % silt / 100 | Hernandez et al., 2012) |

[revised manuscript text omitted]

---

## Referee Report (RR1)

**Review of paper**

**'A review of the (Revised) Universal Soil Loss Equation (R/USLE): with a view to increasing its global applicability and improving soil loss estimates'**

**By R. Benavidez et. al.**

**1. Scope**

The paper provides a thorough introduction into the USLE model family, a group of empirical long term soil erosion models. This paper is of interest to the HESSD community, as the various USLE variants described in this paper are among the most used erosion models overall.

**2. Summary**

The paper gives an introduction into the motivation and method of using USLE models and describes the conceptual background for all individual factors needed to calculate the annual soil loss amounts with USLE models. This is being done by referring to different case studies as well as widely cited papers of variations of USLE models developed to adapt the model to other regions of the world and improve the model family. The calculation formulas of the USLE factors from those papers are provided in tabular form as well, giving a quick overview of these different approaches. The paper also discusses the limitations of USLE models and points at needed future improvements.

**3. General evaluation**

**Scientific significance**

The paper provides a good overview of the topic and goes in depth into the history and motivation of the various USLE models and the possible application use cases of them. This is especially helpful for someone just starting with soil erosion modelling.

**Scientific quality**

The paper is providing a useful overview over the widely used USLE models and their respective equations as well as discussing the limitations of the application of those models. It goes in depth on the problem of validation of modelled results while providing an explicit range of reported under-and over-prediction by the various studies. It mentions the connection of erosion to surface runoff and sediment transport into the rivers and lakes, and the point that this is where the USLE models are lacking and could be improved on.

**Presentation quality**

The paper is structured well, but is lacking in visual descriptions of concepts and equations. Especially a visualization of the equations could make the mathematical concepts behind them more understandable.

**4. Specific comments**

p. 2, l. 5-6: "Understanding and mitigating erosion and associated …" instead of "Managing erosion"

p. 2, l. 28-30: I think this would be a good place to add a mention to the timescale, too, even though it is mentioned a few lines later.

p. 4, l. 25: Sentence seems a bit out of place in this chapter, rather as part of chapter 1?

p.15, l.31: "stream delivery ratio", should be "sediment delivery ratio".

p. 18, l. 12-22: this whole paragraph seems a bit too general for this seasonality section (and a bit redundant). It would be better to move it to the summary and conclusion chapter.

p. 18, l. 23-24: This sentence needs rephrasing in my opinion. Modelling at sub-annual time scale is important **because** of the temporal and spatial variations that are there and if we don't account for them somehow, the model results will be wrong or at least very bad. The understanding of those temporal variations is a prerequisite and not knowledge derived from the application of the USLE model.

p. 19, l. 28: typo: some key  future …

p. 19, summary chapter: Missing a few points that get mentioned during the paper (see remark for p.18, l.12-22).

**5. Additional comments**

I personally think the SDR part is still a little too short, but it would probably be out of scope of the paper to go into more detail.

---

## Editor Decision (ED1)

**Editor decision for manuscript hess-2018-68**

**A review of the (Revised) Universal Soil Loss Equation (R/USLE): with a view to increasing its global applicability and improving soil loss estimates'**

**by R. Benavidez et al.**

Dear Authors,

I have read the referee's comments and your related replies. In general your replies are satisfactory, so please revise your manuscript accordingly and provide both a version with and without track changes.

Please let me emphasize that I agree with the referees that in order to make the manuscript acceptable as a review paper, the following aspects should be considered:

- A critical discussion/evaluation of the R/USLE applications presented in the paper. Have they been applied according to the recommendations, are output uncertainties honestly discussed, etc.
- R/USLE should be put into the greater context of available Soil Loss Models (which models are out there, how do they differ with respect to underlying concepts/simplifications/assumptions, when to use which, data requirements and output quality)
- A critical and honest discussion of R/USLE output quality / predictive uncertainty through comparison with observations and/or other models. I know that related observations are sparse, but this makes collection and comparison even more valuable for a review paper.
- Recommendations on the application of R/USLE,  not only but also with respect to spatial resolution (minimum resolution, recommended resolution, effect of resolution on uncertainty)

Some clarifications of comments by referee #2

- P2 / L6: The key issue here is the word 'transport': Many soil loss models do not explicitly include transport simulations, they just provide erosion rates. So you may want to replace 'by helping understand sediment transport ..' with 'by helping predict erosion rates'
- P6 / L19: The values for summer, winter etc. are independent, so why connect the values from the different studies with lines? Either show as multi-bar plot, or display in a table only.

Yours sincerely,
Uwe Ehret

---

## Author Response (AR2)

Author's Response in Red

**Review of paper**
**'A review of the (Revised) Universal Soil Loss Equation (R/USLE): with a view to increasing its global applicability and improving soil loss estimates'**
**By R. Benavidez et. al.**

**1. Scope**

The paper provides a thorough introduction into the USLE model family, a group of empirical long term soil erosion models. This paper is of interest to the HESSD community, as the various USLE variants described in this paper are among the most used erosion models overall.

**2. Summary**

The paper gives an introduction into the motivation and method of using USLE models and describes the conceptual background for all individual factors needed to calculate the annual soil loss amounts with USLE models. This is being done by referring to different case studies as well as widely cited papers of variations of USLE models developed to adapt the model to other regions of the world and improve the model family. The calculation formulas of the USLE factors from those papers are provided in tabular form as well, giving a quick overview of these different approaches. The paper also discusses the limitations of USLE models and points at needed future improvements.

**3. General evaluation**

**Scientific significance**

The paper provides a good overview of the topic and goes in depth into the history and motivation of the various USLE models and the possible application use cases of them. This is especially helpful for someone just starting with soil erosion modelling.

**Scientific quality**

The paper is providing a useful overview over the widely used USLE models and their respective equations as well as discussing the limitations of the application of those models. It goes in depth on the problem of validation of modelled results while providing an explicit range of reported under-and over-prediction by the various studies. It mentions the connection of erosion to surface runoff and sediment transport into the rivers and lakes, and the point that this is where the USLE models are lacking and could be improved on.

**Presentation quality**

The paper is structured well, but is lacking in visual descriptions of concepts and equations. Especially a visualization of the equations could make the mathematical concepts behind them more understandable.

Agreed, being able to visualise the equations would be a useful component. Most of the RUSLE literature reviewed for this paper lacked any visualisation of their derived equations, making it difficult to understand the relationships between the input values (rainfall, soil texture, etc.) and the resultant sub-factors used in the RUSLE equation. Due to space constraints, it is difficult to put enough meaningful graphs in this broad review.
We will take this suggestion on board in case studies we are currently working on and writing up for publication. In these, fewer equations are presented and we will be able to draw on this paper and others for background. There will be more space to both visualise and more thoroughly explain pertinent sub-factor equations. These case studies will also include maps of subcomponent variations and resultant soil erosion vulnerability under different sub-factor equations so that the reader will be able to better understand how these sub-factors affect soil loss estimates.

**4. Specific comments**

All the specific comments have been taken into account and changes made accordingly.

p. 2, l. 5-6: "Understanding and mitigating erosion and associated …" instead of "Managing erosion"

p. 2, l. 28-30: I think this would be a good place to add a mention to the timescale, too, even though it is mentioned a few lines later.

p. 4, l. 25: Sentence seems a bit out of place in this chapter, rather as part of chapter 1?

p.15, l.31: "stream delivery ratio", should be "sediment delivery ratio".

p. 18, l. 12-22: this whole paragraph seems a bit too general for this seasonality section (and a bit redundant). It would be better to move it to the summary and conclusion chapter.

Agreed, and this paragraph has been shortened and moved to the summary chapter under the section on future work.

p. 18, l. 23-24: This sentence needs rephrasing in my opinion. Modelling at sub-annual time scale is important **because** of the temporal and spatial variations that are there and if we don't account for them somehow, the model results will be wrong or at least very bad. The understanding of those temporal variations is a prerequisite and not knowledge derived from the application of the USLE model.

Agreed, and this paragraph has been slightly overhauled to clarify why sub-annual estimates have an advantage in accuracy over annual estimates.

p. 19, l. 28: typo: some key  future …

p. 19, summary chapter: Missing a few points that get mentioned during the paper (see remark for p.18, l.12-22).

Paragraph originally on p.18, l.12-22 has been shortened and moved to the summary chapter. Additionally, the point about validation and compiling a global database of soil loss estimates for future research has been included in the summary chapter now.

**5. Additional comments**

I personally think the SDR part is still a little too short, but it would probably be out of scope of the paper to go into more detail.

Agreed that this this is out of scope for a broad review paper, but it is a good point that will be addressed in future work. Further work can investigate how the strengths of RUSLE can be combined with SDR for sediment delivery to streams and with the Compound Topographic Index (CTI) for gully erosion. Both SDR and CTI need to be analysed further before combining them with RUSLE, and this could be the scope of a good case study.

[revised manuscript text omitted]

| | | | | | |
|---|---|---|---|---|---|
| | | | | N = number of years | |
| | | | | Units: Megajoule •millimetre • hectare$^{-1}$ • hour$^{-1}$ • year$^{-1}$ | |
| 10 | Fernandez et al. (2003), originally developed by the USDA-ARS (2002) | USA | Annual | $$R = -823.8 + 5.213P$$ P = annual precipitation | USA (Fernandez et al., 2003); Greece (Jahun et al., 2015) |
| | | | | Units: Megajoule •millimetre • hectare$^{-1}$ • hour$^{-1}$ • year$^{-1}$ | |
| 11 | Ram et al. (2004), as cited in Jain and Das (2010) | India | Annual | $$R = 81.5 + 0.38P$$ P = annual precipitation for areas where annual precipitation ranges between 340mm to 3500mm | India (Jain & Das, 2010) |
| | | | | Units: Megajoule •millimetre • hectare$^{-1}$ • hour$^{-1}$ • year$^{-1}$ | |
| 12 | Shamshad et al. (2008) | Malaysia | Monthly and annual | Based on Loureiro and Coutinho (2001) but for Malaysia: $$R = \sum_{i=1}^{12} 6.97 rain_{10} - 11.23 days_{10}$$ $$R = \sum_{i=1}^{12} 0.266 \times rain_{10}^{2.071} \times days_{10}^{-1.367}$$ $$R = \sum_{i=1}^{12} 227 \times \left(\frac{P_i^2}{P}\right)^{0.548}$$ Rain$_{10}$ = monthly rainfall for days with $\geq$ 10.0mm of rain Days$_{10}$ = monthly number of days with rainfall $\geq$ 10.0mm of rain P$_i$ = monthly precipitation P = annual precipitation | Philippines (Delgado & Canters, 2012) |
| | | | | Units: Megajoule •millimetre • hectare$^{-1}$ • hour$^{-1}$ • year$^{-1}$ | |
| 13 | Irvem et al. (2007) | Turkey | Monthly and annual | $$R = 0.1215 \times MFI^{2.2421}$$ $$MFI = \sum_{i=1}^{12} \frac{P_i^2}{P}$$ P$_i$ = monthly precipitation P = annual precipitation | Turkey (Ozsoy et al., 2012) |
| | | | | Units: Megajoule •millimetre • hectare$^{-1}$ • hour$^{-1}$ • year$^{-1}$ | |
| 14 | Ferreira and Panagopolous (2014), similar to | Portugal | Daily | $$R = \sum_{i=1}^{12} 6.56 rain_{10} - 75.09 days_{10}$$ | Portugal (Ferreira & Panagopoulos, 2014) |

| | | | | | |
|---|---|---|---|---|---|
| | Loureiro and Coutinho (2001) | | | $Rain_{10}$ = monthly rainfall for days with $\geq$ 10.0mm of rain
$Days_{10}$ = monthly number of days with rainfall $\geq$ 10.0mm of rain

Units: Megajoule •millimetre • hectare$^{-1}$ • hour$^{-1}$ • year$^{-1}$ | |
| 15 | Nakil (2014) as cited in Nakil and Khire (2016) | India | Annual | $$R = 839.15 \times e^{0.0008P}$$
P = annual precipitation

Units: Megajoule •millimetre • hectare$^{-1}$ • hour$^{-1}$ • year$^{-1}$ | India (Nakil & Khire, 2016) |
| 18 | Naipal et al. (2015) | Global application, but original data from USA and Europe | Annual | Various equations depending on Köppen climate classification, including alternate equations if SDII is not available

P = annual precipitation (mm)
Z = mean elevation (m)
SDII = simple precipitation intensity index (mm day$^{-1}$) | |
| 19 | Klik et al. (2015) | New Zealand | Annual or seasonal | Annual or seasonal:
$$R = aP^b$$
$$R = aP + b$$

P = annual precipitation (mm) or seasonal precipitation (mm)
a & b = constants depending on region of New Zealand

The equation used will depend on the region of New Zealand, and the season.

Units: Megajoule •millimetre • hectare$^{-1}$ • hour$^{-1}$ | |
| 20 | Sholagberu et al. (2016) | Malaysia | Annual | $$R = 0.0003P^{1.771}$$
P = annual precipitation

Units: Megajoule •millimetre • hectare$^{-1}$ • hour$^{-1}$ • year$^{-1}$ | |

**Table 3: Summary of different studies with soil erodibility equations, original locations, and other studies that used their equations. All of the equations in Table 2 use imperial units of soil erodibility: ton • acre • hour • hundreds of acre$^{-1}$ • foot$^{-1}$ • tonf$^{-1}$ • inch$^{-1}$. Multiply by 0.1317 to give in SI units of metric ton • hectare • hour • hectare$^{-1}$ • megajoule$^{-1}$ • millimetre$^{-1}$.**

| # | Author | Original Location | Data requirements | Equation | Other studies |
|---|---|---|---|---|---|
| 1 | Wischmeier and Smith (1978) and | USA | Very fine sand (%), clay (%), silt (%), organic | $$M = Silt \times (100 - Clay)$$ $$K = \{[2.1 \times M^{1.14} \times (10^{-4}) \times (12 - a)] + [3.25 \times (b - 2)] + [2.5 \times (c - 3)]\} \div 100$$ | Thailand (Eiumnoh, 2000); Vanuatu |

| | | | | | |
|---|---|---|---|---|---|
| | Renard et al. (1997) | | matter (%), soil structure, profile-permeability | M = Particle-size parameter
Silt = Silt (%) but also includes the percentage of very fine said (0.1 to 0.05mm)
Clay = Clay (%)
a = Organic matter (%)
b = Soil-structure code used in soil classification:
    • 1: Very fine granular
    • 2: Fine granular
    • 3: Medium or coarse granular
    • 4: Blocky, platy, or massive
c = Profile-permeability class
    • 1: Rapid
    • 2: Moderate to rapid
    • 3: Moderate
    • 4: Slow to moderate
    • 5: Slow

[revised manuscript text omitted]